# Evidence of Tropospheric Uplift into the Stratosphere via the Tropical Western Pacific Cold Trap

Xiaoyu Sun[1], Katrin Müller[1], Mathias Palm[2], Christoph Ritter[2], Denghui Ji[1], Tim Balthasar Röpke[2], and Justus Notholt[1]

[1]Institute of Environmental Physics, University of Bremen, 28359 Bremen, Germany
[2]Alfred Wegener Institute, Helmholtz Centre for Polar and Marine Research, 14473 Potsdam, Germany

**Correspondence:** Xiaoyu Sun (xiaoyu_sun@iup.physik.uni-bremen.de)

**Abstract.**

Understanding air mass sources and transport pathways in the Tropical Western Pacific (TWP) is crucial for determining the origins of atmospheric constituents in the stratosphere. This study uses lidar and balloon observations in Koror, Palau, and trajectory simulations to study the upward transport pathway over the TWP in the upper troposphere and lower stratosphere
(UTLS). During northern hemisphere winter, the region experiences the highest relative humidity and the lowest temperatures (<185 K) at 16–18 km, and is called the "cold trap". These conditions lead to water vapor condensation, forming thin cirrus clouds, which can be measured as an indicator of the ascent of air masses. A representative example from December 2018 shows a subvisible cirrus cloud layer (optical depth < 0.03) measured by lidar, coinciding with high supersaturation observed by radiosonde. Trajectories initiated from the cloud layers confirm that air masses ascend slowly from the troposphere into the
stratosphere primarily during northern hemisphere winter. In contrast, lidar measurements show similar cloud layers during a summer case (August 2022) with warmer temperatures and drier conditions, where air descends after cloud formation, as indicated by the trajectory results. Among all cirrus clouds observed in December and August, 46% of air masses rise above 380 K after cloud formation in December, compared to only 5% in August, possibly influenced by the Asian summer monsoon. These findings underscore the importance of the cold trap in driving air mass transport and water vapor transformations in the
UTLS.

## 1 Introduction

Processes in the tropical upper troposphere and lower stratosphere (UTLS) have been known to modulate the global climate strongly for more than 60 years. Many studies have investigated this region since the first observations by Brewer (1949). During the Northern Hemispheric (NH) winter (December–February, DJF), tropospheric air ascends into the stratosphere via
the Tropical Tropopause Layer (TTL, approximately 14–18 km), primarily in the Tropical Western Pacific (TWP) (Fueglistaler et al., 2004; Rex et al., 2014; Bourguet and Linz, 2023). The TWP extends from the Maritime Continent to the International Date Line and features some of the world's highest sea surface temperatures. The TWP warm pool provides an environment conducive to the development of deep convective cloud systems (Fueglistaler et al., 2004, 2009). Deep convection and large-

scale ascent in this region enable boundary layer air, heated by the warm ocean surface, to ascend to the TTL, altering the

composition of the TTL atmosphere across the tropics. In NH summer, the strong deep convection within the Asian summer monsoon (ASM) becomes an important regional counterpart of the TWP for the uplift of air masses into the UTLS (Fueglistaler et al., 2004; Randel et al., 2010; Pan et al., 2022). Furthermore on a subseasonal scale, eastward shedding events of the ASM anticyclone can directly affect the TWP by amplifying the western Pacific anticyclone and enabling transport of ASM processed air masses even into the deep tropics (Honomichl and Pan, 2020a; Pan et al., 2022; Vogel et al., 2019).

During NH winter, the TTL over the TWP exhibits some of the coldest temperatures in the UTLS (Fueglistaler et al., 2009). Relative humidity with respect to ice (RHi) in this layer can be near or above saturation (Jensen et al., 2005), enabling the formation of thin cirrus clouds (Gettelman and de Forster, 2002; Bourguet and Linz, 2023). The region with the lowest temperature, which is called the cold trap (Holton and Gettelman, 2001), significantly influences the formation of cirrus clouds and thus the humidity in the TTL by trapping for moisture (Schoeberl et al., 2019). As air masses move quasi-horizontally within

the TTL for long distances, they slow ascent, experiencing the coldest temperatures (Li et al., 2020; Hasebe et al., 2013). Moisture is removed via condensation in a freeze-drying and subsequent sedimentation process(Ueyama et al., 2014). The air mass experiences the lowest temperature as well as the water vapor minimum before it leaves the TTL and eventually enters the stratosphere over the TWP (Fueglistaler et al., 2004; Rex et al., 2014). This has been confirmed with model calculations in previous studies where Lagrangian trajectories were calculated (Fueglistaler et al., 2004; Holton and Gettelman, 2001; Pan

et al., 2019). Observations in the TWP are important to reconcile the debates about the TTL air mass uplift with the "stratospheric fountain" hypothesis (Newell and Gould-Stewart, 1981; Sherwood, 2000). Under global warming, it has been reported that the weakening of the cold trap is related to the increased Brewer-Dobson circulation (BDC) strength (Fueglistaler et al., 2014; Fu et al., 2019; Bourguet and Linz, 2023). Studies have reported an increase in the upward motion of the tropospheric air over the TWP, which means that more tropospheric trace gases and aerosols may enter the stratosphere through this region,

affecting stratospheric chemistry and climate (Qie et al., 2022). Since stratospheric water vapor is an important greenhouse gas, the weakening of the cold trap is predicted to have positive feedback on the greenhouse gas forcing of surface warming (Keeble et al., 2021).

The cold trap in the TTL has a major influence on global climate through several processes, including the high-altitude cirrus clouds. Due to their cold temperature and low optical thickness, these thin clouds have a low albedo but a greenhouse effect

that can modulate the radiative budget of tropical climate systems (Woods et al., 2018; Taylor et al., 2011). There have been discussions suggesting that high-altitude subvisible cirrus (SVC), with a cloud optical depth (COD) of less than 0.03 (Sassen and Cho, 1992), and extremely thin tropical cirrus (ETTCi) (Immler et al., 2007; Sun et al., 2024b) in the TTL, are often associated with freeze-drying process and thus the dehydration of air. This dehydration typically occurs before the air enters the stratosphere, e.g. Jensen et al. (1996). However, SVC and ETTCi and its interaction with water vapor in the TTL are still

not fully understood. This is due to the lack of direct observations in the tropical oceanic regions specifically the TWP, where SVC/ETTCi are prevalent (Cairo et al., 2021). The Space-borne Cloud-Aerosol lidar with Orthogonal Polarization (CALIOP) aboard Cloud-Aerosol Lidar and Infrared Pathfinder Satellite Observations (CALIPSO) partially filled the observational gap of cirrus clouds in the last 17 years (Bertrand et al., 2024). However, because of the narrow swath of CALIOP, the repeat cycle

is only around 16 days in the tropical regions (Pandit et al., 2015). The continuous measurements of cirrus clouds by well-calibrated ground-based observations with lidar instruments in the tropics are essential for understanding the cloud formation and related processes as dehydration and associated implications for stratosphere-troposphere exchange (STE).

The observational gap in the TWP has improved since the establishment of the Palau Atmospheric Observatory (PAO) in the western Pacific warm pool in Palau (7.3°N, 134.5°E) since 2016 (Müller et al., 2024a; Sun et al., 2024b). The station is equipped with a ground-based solar absorption Fourier transform infrared spectrometer (FTIR), a ground-based COMpact Cloud and Aerosol Lidar (ComCAL) (Immler et al., 2007; Sun et al., 2024b) since 2018 and balloon sounding systems with ozone- ($O_3$), water vapor- ($H_2O$), aerosol- and radiosondes (Müller et al., 2024a). The long-term monitoring of high cirrus clouds above Palau can provide insights into STE in this region, with SVC and ETTCi being specifically important indicators. This study was motivated by the need to study dynamic transport paths over the TWP using measurements of high cirrus cloud components in combination with trajectory analysis. The cloud measurements we used are from December 2018 and August 2022, representing the NH winter and summer seasons, respectively. From our long-term measurements since 2018 in Palau, there is no significant yearly difference in the cirrus cloud measurements (Sun et al., 2024b). In August 2022, we increased all operations at the PAO within the Asian Summer Monsoon Chemical & Climate Impact Project (ACCLIP, https://espo.nasa.gov/acclip/content/ACCLIP), which particularly targeted the influence of the ASM on the Western Pacific region using aircraft measurements, conducted from an airbase in the Republic of Korea (Pan et al., 2024). Our coordinated measurements with several other ground stations alongside the comprehensive aircraft observations allow detailed studies of the chemical composition along transport pathways, which will be the subject of future studies.

In Sect. 2, we present the observational data used in this study and the setup of the trajectory model simulations by the Hybrid Single-Particle Lagrangian Integrated Trajectory model (HYSPLIT) and Alfred-Wegener-InsTitute LAgrangian Chemistry/Transport System (ATLAS). We present the two case studies in winter and summer, the trajectory simulation results, and a schematic illustration of the dynamic transport pathway above the TWP (Sect. 3). In the discussion (Sect. 4), we discuss the possible effects of changes in transport pathways on stratospheric composition in the context of seasonal variations in pathways and the origin of trace gas components in tropospheric air masses. In Sect. 5, conclusions are drawn.

## 2 Methods and Data

### 2.1 Cirrus Cloud Lidar Observations

The cirrus cloud measurements are obtained from the ground-based Compact Cloud and Aerosol Lidar (ComCAL) from 04/2018-08/2022 conducted at the Palau Atmospheric Observatory (PAO) located in the center of the tropical Western Pacific warm pool (Müller et al., 2024a; Sun et al., 2024b). Ground-based lidar measurements are suitable and practical to continuously monitor particles, namely cloud layers and aerosols, and it has been used in many previous studies for the analysis of cirrus clouds in the tropics (Cairo et al., 2024; Pandit et al., 2015; Voudouri et al., 2020). The lidar system has been operated since April 2018 and continued measuring until May 2019. Due to technical issues, no measurements were performed between May 2019 and March 2022. After maintenance, ComCAL restarted operations from March 2022 until December 2022. In this study,

**Table 1.** Overview of the ComCAL observations and cirrus cloud geometrical properties in December 2018 and August 2022 during ACCLIP in Palau.

| | December 2018 | August 2022 |
|---|---|---|
| Total lidar sampling time (h) | 22.3 | 113.8 |
| The occurrence time of cirrus clouds (h) | 11.3 | 63.8 |
| The percentage occurrence (PO) of cirrus clouds (%) | 50.6 | 47.7 |
| Monthly mean mid-cloud temperature (°C) | $-78.1 \pm 7.1$ | $-69.9 \pm 9.0$ |
| Monthly mean cloud base height (km) | $15.3 \pm 1.4$ | $13.9 \pm 1.8$ |
| Monthly mean cloud top height (km) | $16.9 \pm 1.0$ | $15.8 \pm 1.2$ |

The details of the ComCAL data processing and the method to calculate the cloud properties are described in Sun et al. (2024b).

most of the results are based on the observations obtained in two months: December 2018 and August 2022. These two months represent two distinct seasonal features of local cloud layers (Sun et al., 2024b). The primary geometrical properties of the cirrus clouds observed over Palau are summarized in Table 1. The cloud base and top heights are determined individually for each cloud layer. The mid-cloud temperature is defined as the temperature at the midpoint of each cloud layer, calculated as the average of the cloud top and base heights. If more than one cloud layer is observed simultaneously, they are considered as separate cloud layers. Detailed discussions on quantifying cirrus cloud properties such as cloud layer heights, temperature, and the treatment of multi-layer clouds are given in (Sun et al., 2024b). Table 1 demonstrates that the selected cases from December and August effectively represent typical cirrus cloud conditions observed during these two distinct seasons over Palau. The mid-cloud temperature in December is generally lower, and the cloud layers are higher compared to August, reflecting seasonal variations in meteorological conditions associated with the cold trap. These seasonal differences will be further described in Sect. 3.1, supporting reasons for selecting these two months for the following analysis.

## 2.2 Radiosonde Observations

The temperature and moisture profiles based on the analysis for meteorological conditions over the TWP are from radiosonde observations conducted at the Palau Weather Service (PWS, station reference number: PTRO 91408). Radiosondes are launched twice daily, at 0:00 and 12:00 UTC. The launch site was located close to the PAO until August 2018, before it was placed at the international airport, approximately 9.5 km away. The pressure, temperature, potential temperature, and relative humidity (RH) profiles from the radiosonde closest in time to the lidar observations are used.

## 2.3 Trajectory Simulations

To investigate the history and future of cirrus cloud air masses over Palau, we conducted trajectory analysis using the Air Resources Laboratory's (ARL) Hybrid Single-Particle Lagrangian Integrated Trajectory model (HYSPLIT) (Stein et al., 2015). In this study, meteorological data for HYSPLIT were selected from the operational system Global Data Assimilation System

(GDAS, Kanamitsu (1989)) (1° × 1°, 3-h temporal resolution) from the National Centers for Environmental Prediction (NCEP) (National Centers for Environmental Prediction, National Weather Service, NOAA, U.S. Department of Commerce, 2000). GDAS/NCEP is commonly used for air transport simulations by HYSPLIT (Stein et al., 2015).

This study uses the HYSPLIT model with its kinematic approach to the vertical displacement of air masses to calculate the backward and forward trajectories. The trajectories are initiated from the cirrus cloud air masses with cloud top and base height $C_{base}$ and $C_{top}$ and corresponding measurement time from ComCAL observations. Within each cirrus cloud layer, five trajectories are released at the same point in time, but evenly distributed by altitude. These 5 trajectories are started every hour, based on the assumption that within 1 h, the variation of the cirrus cloud can be neglected. This assumption is adapted from Cairo et al. (2021), which used a time step of 3 h in the trajectory analysis of the cirrus cloud above Palau using a similar gound-based lidar system. HYSPLIT does not specifically account for localized convective processes that contribute to convective clouds (Loughner et al., 2021). When these clouds are detected, it suggests that condensation processes have occurred, potentially leading to associated dehydration of air masses through the removal of water vapor by ice particle formation and their subsequent sedimentation.

To investigate the sensitivity of trajectory pathways calculated with HYSPLIT setup, we conducted additional simulations using the transport module of the Lagrangian Chemistry and Transport model (Alfred-Wegener-InsTitute LAgrangian Chemistry/Transport System (ATLAS) (Wohltmann and Rex, 2009), driven by the European Centre for Medium-Range Weather Forecasts (ECMWF) fifth-generation reanalysis dataset (ERA5; Hersbach et al. (2020)) with a hybrid vertical coordinate that gradually changes from kinematic to fully diabatic above 100 hPa. In this study, a thorough inter-comparison between the two models is not the focus. Instead, the relative distributions of different pathways are of main interest. In this sense, the example case study results between the two models are given, and the differences in reanalysis data inputs are neglected. For details about the ATLAS model setup and exemplary simulation results, please see Appendix A.

## 3   Results

### 3.1   Climatology of the Western Pacific Cold Trap

Figures 1a and b show the cold point tropopause (CPT) in the tropical region for two reanalysis datasets, ERA5 (color scales) and NCEP (contour lines). Both data sets show similar seasonal features above the TWP. There is a lower tropopause potential temperature during December-February (DJF) compared to June-August (JJA) above the TWP, indicating a minimum temperature and humidity, which are associated with the cold trap and its dehydration processes. The cold point is weakly confined above the TWP during summer months, as shown in Fig. 1b. Instead, the influence of the ASM on the NH tropopause altitude is apparent.

The longitudinal height-cross-section of the mean temperature over the tropical region (30°S–30°N) is shown in Figs. 1c and d. The TWP and the Palau site (marked by the white dashed line) are located within a cold trap structure with the lowest temperatures from 120 hPa (16 km) to 90 hPa (18 km) and a minimum center at approximately 100 hPa. The climatology of tropopause temperatures exhibits a seasonal cycle, with minimum temperatures below 192 K during December-February

and around 198 K during June-August (Fig. 1c-d). Correspondingly, the spatial extent of the cold centers is more extensive in NH winter than in summer, following the same seasonal pattern. Additionally, the cold trap above the TWP reaches higher geometric heights during December-February, extending above 70 hPa (18.5 km). This feature is associated with the highest geometrical level of the isentropes, e.g., $\theta$ = 360, 370, and 380 K in Fig. 1c, above the TWP, compared to elsewhere in the tropics.

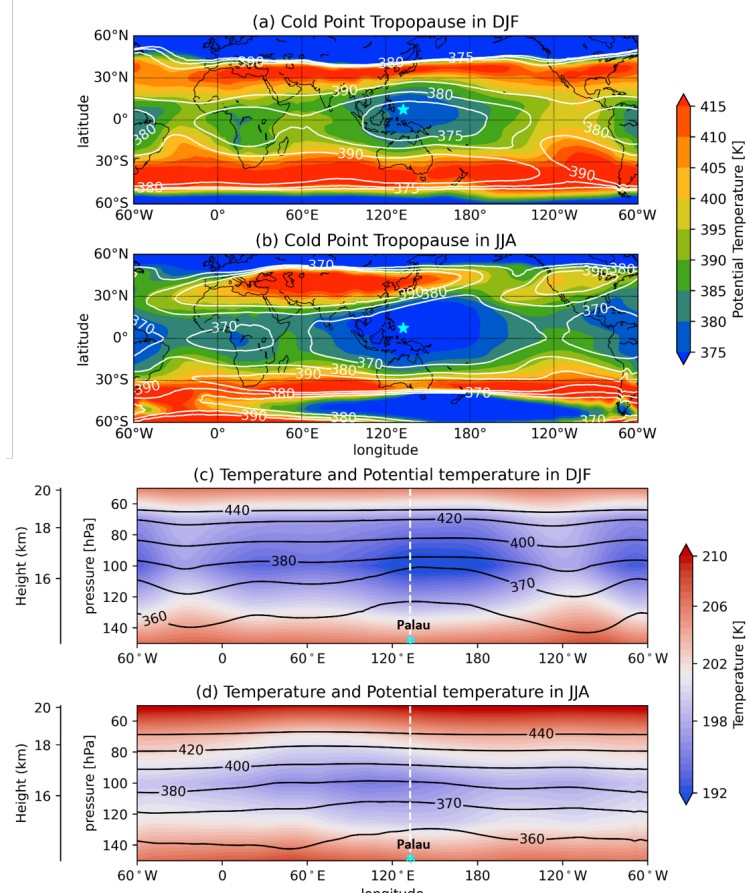

**Figure 1.** (a) and (b): Climatologies of tropopause (cold point) potential temperatures (in Kelvin, K) calculated from ERA5 (1980 to 2019, color scales) and NCEP-NCAR (1980–2019, contour lines) for December, January, and February (DJF, for the NH winter) and June, July, and August (JJA, for the NH summer), respectively. (c) and (d): Climatologies of pressure/height - longitude cross sections of temperature (shading) and potential temperature (contours) zonally averaged between 30°S - 30°N from ERA5 (1980-2022) for DJF and JJA, respectively. The cyan marker in (a) and (b) shows the location of the Palau site (7.3°N, 134.5°E), which is located below the cold trap. The white dashed line in (c) and (d) shows the longitude of the Palau site. The tropopause data in plot (a) and (d) are from the Reanalysis Tropopause Data Repository (Hoffmann and Spang, 2022; Zou et al., 2023; Hoffmann and Spang, 2021). The temperature and the potential temperature in (c) and (d) are from the ERA5 reanalysis data (Hersbach et al., 2020).

## 3.2 Case Studies

We analyzed two typical cases of cirrus cloud layers measured in December and August, combining meteorological conditions including temperature and RH profiles obtained from the radiosonde, measurements of cirrus clouds in the tropical tropopause layer (TTL), and trajectory simulation results.

The first case, on 13 December 2018 (Fig. 2), depicts a typical December situation measured above Palau with two distinct layers of clouds. During the measurement period, an SVC (COD < 0.03) and an ETTCi (COD < 0.005) are detected at approximately 18 km, and a thin cirrus is detected at 14.5 to 16 km, as shown in Fig. 2a. The COD of each cloud layer at the time of measurement is presented in Fig. 2b, including measurement gaps of the ComCAL lidar.

     The radiosonde was launched at 12:00 UTC, and Fig. 2c and d present the RHi and temperature profiles, respectively. The

height of the CPT is about 18 km, with a cold point temperature of about 185 K. The CPT measured by radiosonde above Palau on 13 December 2018 agrees with the climatological cold trap at 70 - 100 hPa (16.5 - 18.5 km) above the TWP (Fig. 1b). When TTL air masses are advected horizontally through large-scale circulation into the cold trap, the low-temperature conditions cause water vapor to condense and form cirrus clouds, which in turn indicate vertical transport into the stratosphere. The RHi increases from approximately 14.5 km, which is also the base of the lower cloud layer. The maximum RHi reaches

approximately 160% at 18 km. This supersaturated and cold environment near the CPT suggests a condition favorable for cirrus cloud formation and further dehydration of the air mass via ice particle sedimentation in the altitude range above 14 km up to the CPT.

     The sub-saturated environment measured by the radiosonde within the entire lower cloud altitude range suggests that the cloud measured by ComCAL is dissolving. Despite that, the lidar continuously detected the cloud for three hours. This sug-

gests a continuous supply of water vapor potentially advected from an active convective cell close by or from colder regions elsewhere, which may be an effective mechanism for the local deposition of water vapor. Krämer et al. (2020) reported low RHi in convective overshoots in the TTL and their significant local contribution of water vapor as well, corresponding with our case study for the cloud layer with sub-saturated environment measured by the radiosonde (Fig. 2c). However, the exact mechanism and its impact need further investigation to fully understand the water vapor advection process. Another hypothet-

ical explanation is that the radiosonde was drifting during the ascent and measured outside the cloud, i.e. radiosonde and lidar did not sample the same air mass. However, since these thin cirrus cloud structures often have a large horizontal extent and the lidar measurement is fairly constant over a longer period, this explanation is unlikely.

     To further analyze the path of STE with different cloud types on this day, 20-day backward and forward trajectories, initiated in the upper and lower cloud layers are presented in Fig. 3. For long-lasting cirrus clouds, one back or forward trajectory is

initiated every hour to track their variations over time. The trajectory setup details are described in Sect. 2.3 and Appendix A. The absolute height of the cloud layers and thus trajectory starting points can be inferred from Fig. 2a. For the backward trajectories (Fig. 3a and c) of the upper cloud, the long-distance trajectories from lower altitudes in the tropics are dominant. Several upper cloud layer backward trajectories (Fig. 3a) at high altitudes (17 - 18 km) exhibit circling patterns inside the cold

trap, where the height of the trajectory points remains almost the same as the initial points of the trajectory. In contrast, most backward trajectories for the lower cloud layer originate from long distances at lower altitudes ($\leq 14$ km) as shown in Fig. 3c.

The forward trajectories, shown in Fig. 3b and d, ascend further up from the upper cloud layer (approximately 18 km). Thus, air parcels have traveled long distances in the TTL before reaching the cold trap and eventually entering the stratosphere above the TWP. For the forward trajectories of the lower cloud layer, the situation is similar to that of the upper cloud layer. Almost only within the cold trap region, the height of the trajectory points is higher than the initial point ($\geq 16$ km), while those outside the cold trap tend to remain at lower altitudes. Note that at the end of the forward trajectory, nearly 20 days after initialization, there is a strong updraft above 19 km (Fig. 3b). These unrealistic results may be due to the larger dispersion of the vertical coordinates of stratospheric kinematics by HYSPLIT, as kinematic trajectories are known to be more dispersive compared to the diabatic approach (Bourguet and Linz, 2022). We compared and validated the HYSPLIT results with those of ATLAS. In comparison, ATLAS trajectories do not show this extreme uplift. Otherwise, the results of the two models are comparably consistent, especially regarding the NH winter air uplift. A detailed comparison between the two model results is described in Appendix A.

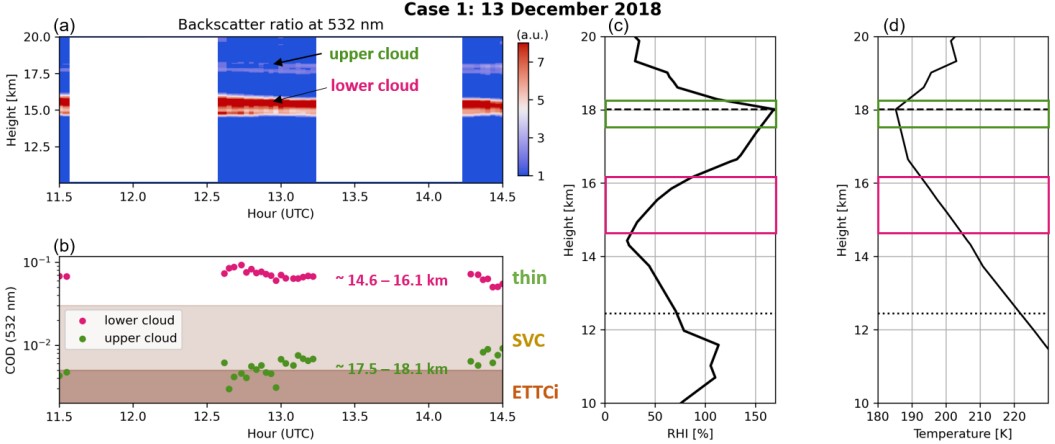

**Figure 2.** Lidar and radiosonde observations for a typical case of cirrus cloud measurements on 13 December 2018. (a) The backscatter ratio (BSR) at 532 nm, thin cirrus (COD < 0.3), SVC (Sub-visible cirrus, COD < 0.03) and ETTCi (extremely thin cirrus, COD < 0.005) are marked. . (b) The cloud optical depth (COD) at 532 nm of the upper and lower cloud layers as a function of time corresponds to Fig. 3a. (c) The relative humidity with respect to ice (RHi) profiles as a function of height. (d) the temperature profiles as a function of height. The temperature and RHi profiles are obtained from the radiosondes launched at the Palau weather station (see Sect. 2). The horizontal lines indicate the altitude range of the TTL between the CPT height (dashed line, upper boundary of the TTL) and the height of LMS (level of minimum stability, dotted line, lower boundary of the TTL) calculated by the temperature profiles each day. Two cloud layers are marked by the box in the profile plot (c) and (d), with the same color code as in (b): the lower cloud marked as magenta and the upper cloud marked as green.

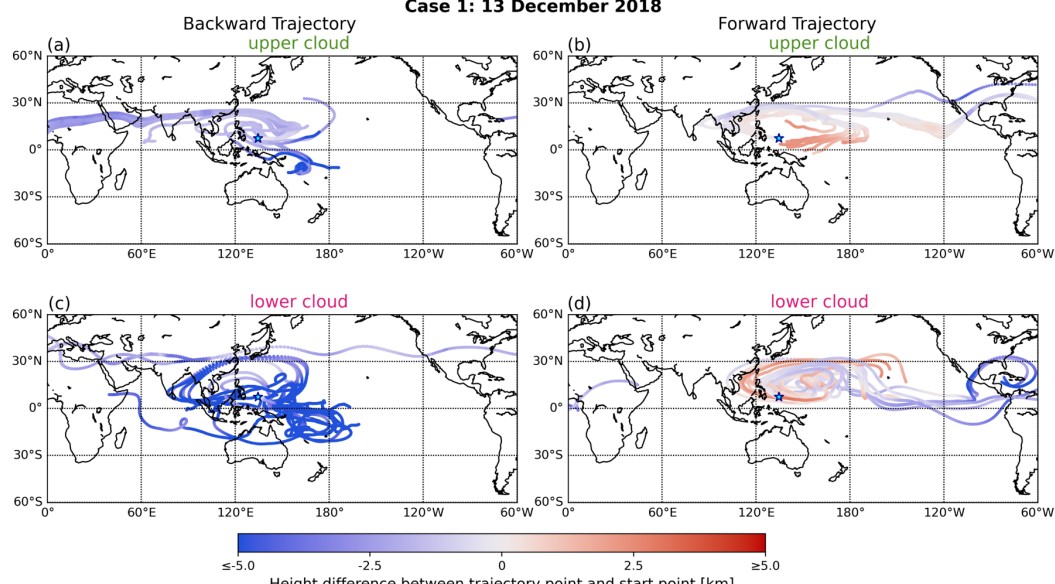

**Figure 3.** 20-d trajectories of the upper cloud (17.5–18.1 km, a–b), and the lower cloud (14.6–16.1 km c–d) for the winter case (Case 1). The left and right columns show the backward (a and c) and forward (b and d) trajectory points released from the cloud layers, respectively. The color scale of the trajectory point scatter depicts the height difference between the trajectory points and the initialization. For the plot with the color scale with the actual height of the trajectory points, please see Fig. S1 and S2. The starting point of the trajectory at Palau is marked by the blue star. Trajectory points are output at hourly intervals but sparsified at intervals of 5 points for the clarity of display.

Another case observed on 1 August 2022 during the ACCLIP campaign, is presented in Fig. 4. During the measurement period, two layers of cirrus were detected, similar to the case on 13 December 2018. The lower cloud layer was located at approximately 14 km with a geometrical thickness of 1 km, and most of the COD of the lower cloud layer is higher than 0.1 200 or even 0.3, indicating thin and thick cirrus, respectively. The upper cloud layer stays below the CPT of about 17.5 km during the measurement period, as shown in Fig. 4d. The COD of this layer was less than 0.03 or even 0.005 at specific measurement times. Thus, the upper layer cloud is classified as SVC and ETTCi, similar to the case on 13 December 2018.

The temperature and RHi are presented in Fig. 4c and d, respectively. The RHi was less than 100% throughout the vertical range from 10 km to 20 km, indicating a non-supersaturated environment compared to the winter case. Even below the cloud 205 bases of both cloud layers, the RHi was less than 100%, which is a sub-saturation condition lower than the upper cloud layer of the winter case (Fig. 2c). However, as mentioned before, the lidar continuously detected the cloud for hours, which suggests a potentially continuous supply of water vapor. Further study is still needed to understand the cold trap in influencing air mass transport and water vapor transformations in the UTLS. The cold point in Case 2 (summer) was about 190 K which is higher than in Case 1 (winter), indicating the less intense cold trap over the TWP. The lower and sub-saturated RHi below the CPT 210 suggests that the cold trap during this period is not cold enough to further dehydrate the air parcels passing through it. Cloud formation in the cold trap releases latent heat, which may influence local temperature and stability. However, the radiative

effects of cirrus clouds can either amplify or suppress vertical motion, depending on ice crystal shape and size (Spang et al., 2024). Specifically, ultrathin cirrus clouds composed of spherical ice crystals tend to warm the atmosphere, while aggregated or hexagonal ice crystals lead to cooling. These cloud microphysical effects interact with large-scale circulation, potentially influencing the vertical transport patterns.

The trajectory results for this case are shown in Fig. 5. Most of the backward trajectories of the cloud layers are from lower heights than the initial point and have travelled long distances, which suggests ascending air masses from the low troposphere, $\leq 15$ km (Fig. 5a) and $\leq 11$ km (Fig. 5c) for the upper and lower cloud, respectively. This August pattern is in contrast to stable circulating air masses in the high TTL of about 17–18 km in the December case (Fig. 3b and d). For the lower cloud, one of two pathways shown by backward trajectories can be related to the active ASM anticyclone in the upper troposphere and its particular northeastward expansion in 2022, combined with the westward extension of the Western Pacific Subtropical High (WPSH) at lower altitudes (Pan et al., 2024) (dominant pathway in ATLAS calculations, see Fig. A3c). ATLAS calculations show an ASM influence still for lower altitudes of the upper cloud, while confirming the dominant long-distance origin of air masses coming from the East (Fig. A3a).

The forward trajectories indicate that almost all the air masses descend from the cloud layer height of approximately 17 - 18 km to lower altitudes below 16 km as shown in Fig. 5b. The trajectories and measurements on 1 August 2022 suggest that although SVC and ETTCi formed near the CPT in the low-temperature layer, they do not indicate a stratospheric pathway. Both clouds rather dissipate due to the less intensive cold trap and the associated non-supersaturated environment, with air masses slowly descending and eventually entering the mid-troposphere.

Note that some air over the equatorial Indian Ocean and Africa ascends higher than 18 km (see Fig. 5b). This feature has not been captured by the ATLAS simulation (Fig. A3b) and is somewhat contradictory to the ongoing negative phase of the Indian Ocean Dipole at the time, which usually suppresses convective activity at the equatorial Eastern coast of Africa. As mentioned before, the intense uplift in the last several days of the trajectories modeled by HYSPLIT may be due to the larger dispersion in stratospheric trajectory calculations (Bourguet and Linz, 2022). Instead, this particular uplift might be related to an equatorial Kelvin wave induced by the western Pacific monsoon (WPM) convection active this time of the year, causing a low temperature layer (Kikuchi and Takayabu, 2004; Immler et al., 2008). The high-low-temperature anomaly wave results in the freezing of the liquid droplets forming optically thin cirrus detected by the lidar measurements (Case 2, Fig. 4b). The WPM is characterized by intense convective activity during the NH summer, generating equatorial Kelvin waves through the release of latent heat from deep convection (Das et al., 2022). These waves propagate eastward, inducing alternating high- and low-temperature anomalies in the TTL. The cooling phases of these anomalies facilitate the freezing of supercooled droplets, leading to the formation of optically thin cirrus clouds, as detected by ComCAL in this study over Palau and previous studies over other regions (Sun et al., 2024b; Immler et al., 2008).

These two cases from two seasons indicate two different pathways of air mass transport above the TWP. A comparison of observed SVC and ETTCi and the associated trajectories shows that clouds with similar optical depth rather ascend in winter and descend in summer. This contrast aligns with the seasonal variability of the BDC, which is stronger in NH winter, supporting enhanced upwelling over the TWP due to an extremely cold "cold trap". In contrast, during NH summer, the BDC

is weaker over Palau, and no evident large-scale upward motion in the trajectory analysis is found. This implies that seasonal differences in the strength of the BDC likely play a more important role than cloud microphysical processes in modulating transport pathways in the TTL. This BDC-driven seasonal pattern is reflected in the December and August cases.

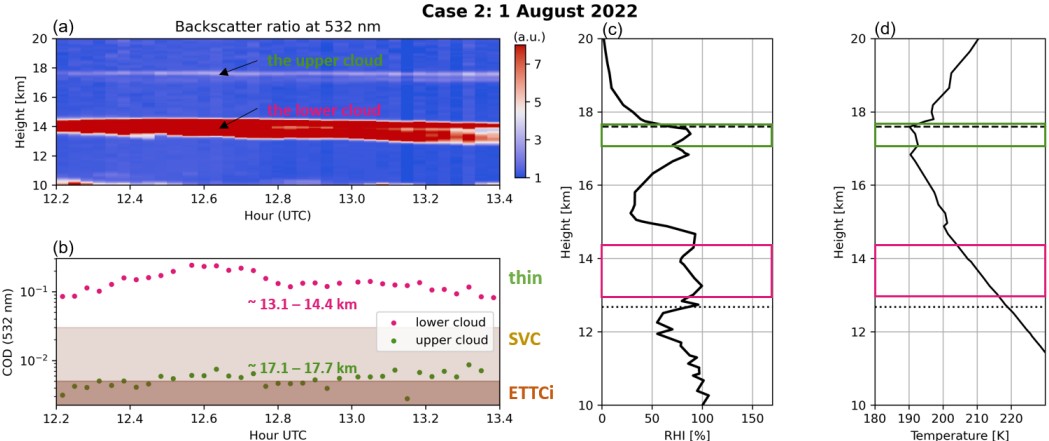

**Figure 4.** Lidar and radiosonde observations of a typical case of cirrus cloud measurements on 1 August 2022. The description of the plots is the same as in Fig. 2.

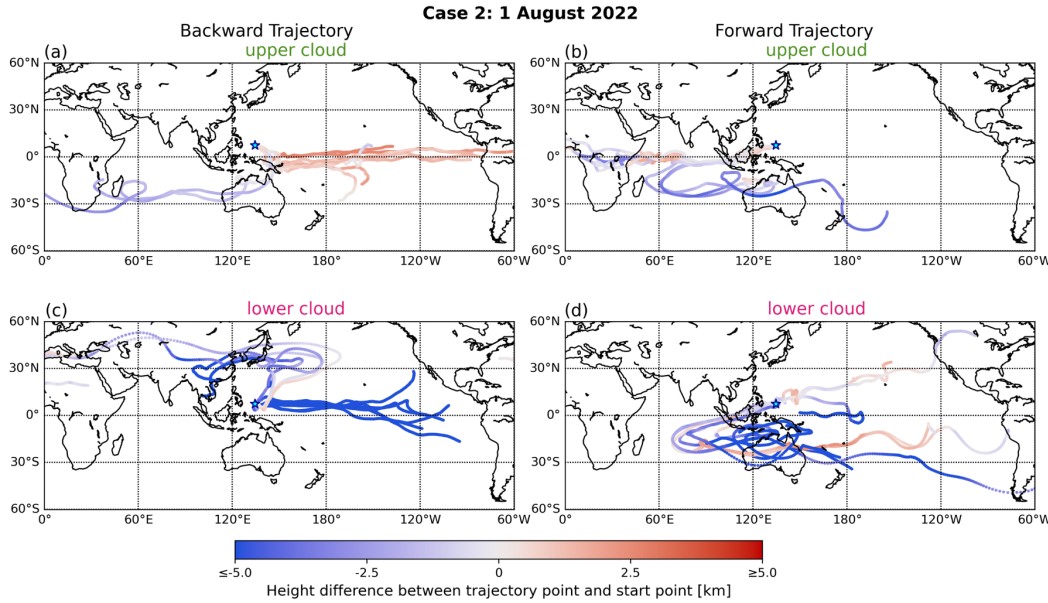

**Figure 5.** 20-d trajectories of the upper cloud (17.1–17.7 km, a–b), and the lower cloud (13.4–14.4 km, c–d) for the summer case (Case 2). The description of the plots is the same as in Fig. 3.

## 3.3 Winter and Summer Monthly Results

In the previous section, we discussed the pathways into the stratosphere during NH winter and compared them to summer by presenting two case studies from these two seasons. To further investigate the transport pathways of the air masses in the cold trap after cirrus cloud formation, we performed trajectory analyses for all cirrus clouds measured by ComCAL in December 2018 and August 2022. Additionally, we present the potential temperature as an additional meteorological parameter along the previously calculated trajectories to see the quasi-horizontal transport over the TWP. As for the case studies, the trajectories are initiated from the time and altitude of cirrus clouds measured above Palau. Within the cloud layers detected by lidar, multiple trajectories are used to capture variations in time and vertical position of the air masses (see Sect. 2.3 and Appendix A). Although the output time step of the trajectory is 1 hour, points are plotted here at daily intervals (every 24 hours) to enhance visual clarity. All trajectories were originally calculated over 20 days, but only the first 10 days are presented here to avoid overwhelming the figure; full 20-day trajectories are shown in Fig. S3 in the supplement. The potential temperature of the starting points ($\theta_0$) is used to distinguish the vertical levels of the cirrus clouds, by three levels with 10 K as interval, i.e., $\theta_0 > 370$ K, 360 K $< \theta_0 < 370$ K, and $\theta_0 < 360$ K. The results are shown in Fig. 6 which shows the potential temperatures of the air along the trajectory.

The monthly transport pattern is consistent with the case studies, with air ascending to a higher level in December and descending to lower levels in August. In the forward trajectories for December, air with $\theta_0 > 370$ K shows a clear ascent toward high levels (Fig. 6a). Air in the upper TTL enters the stratosphere via the cold trap above the TWP. Air masses with 360 K $< \theta_0 < 370$ K, mostly travel to higher levels above Southeast Asia and the TWP but descend over the central and east Pacific in December (Fig. 6c). In contrast, in August, most of the air masses descend, only apart from air masses above the inner tropical Indian Ocean. This latter circulation pattern is likely related to the ASM, similar to the 1 August case study, shown by the trajectory points with $\theta > 370$ K in the equatorial 60°E - 120°E range in Fig. 6b.

In the lower layer ($\theta_0 < 360$ K), trajectories display the greatest horizontal dispersion, influenced by regional atmospheric circulations closer to the lower troposphere. December trajectories remain relatively confined near the Indian Ocean and adjacent regions, while August trajectories spread farther eastward.

## 3.4 The Air Mass Entry Fraction

To quantify the seasonal differences in the transport pathways above the TWP, particularly the dominant transport of air to the stratosphere in the cold trap during December compared to August, we introduce an air mass entry fraction (AEF) based on the forward trajectory analysis and lidar measurements of cirrus clouds. This way, we estimate how much air enters different geographical regions and vertical levels. A schematic illustration of the AEF is shown in Fig. 7. The AEF (%) is calculated as follows:

$$\text{AEF} = \text{N}(\text{x}, \theta) \Big/ \sum \text{N} \times 100\% \tag{1}$$

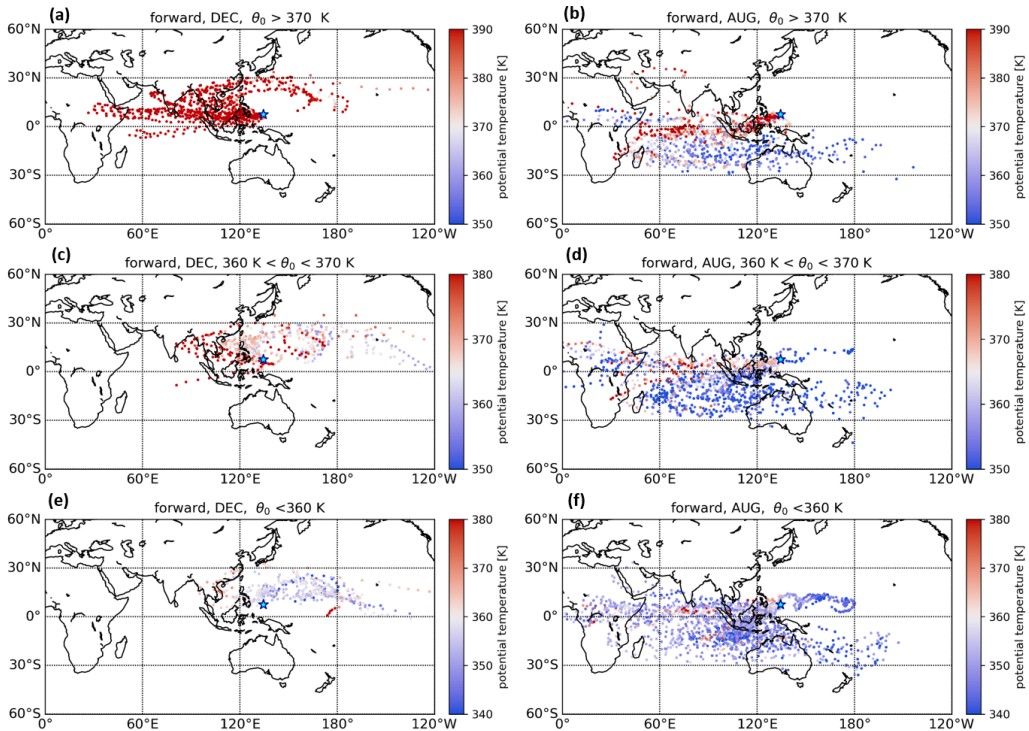

**Figure 6.** 10-d Forward trajectories initiated from the cloud layer in December 2018 (left column, a, c, e) and August 2022 (right column, b, d, f). The trajectories are divided by potential temperature of the starting point ($\theta_0$) into three layers presented in three rows: $\theta_0 > 370$ K (a and b), 360 K $< \theta_0 <$ 370 K (c and d), $\theta_0 <$ 360 K (e and f). The potential temperatures of the trajectory points are given in color. Note the changes in color scales for different layers. Trajectories are initialized at cirrus cloud layer measurements of ComCAL at the PAO, shown by the cyan marker.

where $N(x, \theta)$ represents the total number of trajectory points in the specified box region. Here, x corresponds to different geographic regions, and $\theta$ represents potential temperature, ranging from $\leq 350$ K to $\geq 370$ K in 10 K intervals (Fig. 7a). For instance, N (1, $\theta \geq 400$ K) represents the number of trajectory points in the central east Pacific region within the vertical layer above 400 K (as indicated by the dark green box in Fig. 7b. The AEF is then calculated as the fraction of trajectory points in each box region to the total number of trajectory points. The regional definitions (box regions 1–6) used here follow Sun et al. (2023a), adapted from Fueglistaler et al. (2004) are restricted to ±30° latitude to maintain a tropical focus.

Figure 8 presents the results of the AEF for December and August. The box regions with high AEF values are concentrated above the TWP and Indian Ocean regions in both months. The seasonal differences in the air masses entering the level above 380 K are evident. In December, 46% of the air masses reach this level, whereas in August, only 5% do so. The significantly higher AEF in December (46%) reaching 380 K is consistent with stronger upwelling over the TWP during NH winter. This aligns with the seasonal phase of the Brewer-Dobson Circulation, which facilitates upward transport of TTL air into the

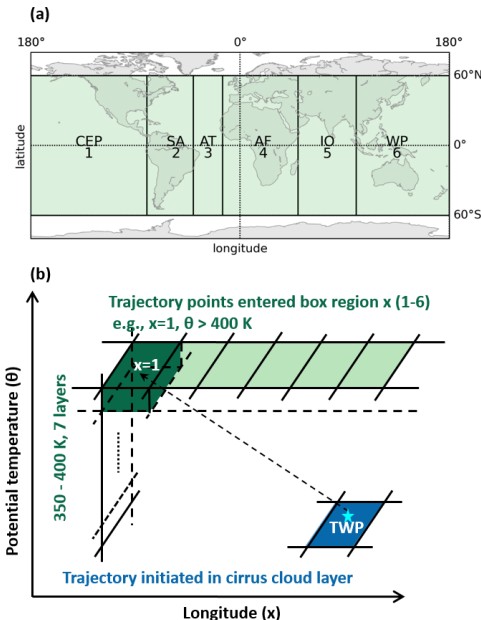

**Figure 7.** Schematic illustrations of the air mass entry fraction (AEF) of the trajectory points. (a) The geometrical definition of each region x: 1: Central & East Pacific (CEP, 180°, 80°W); 2: South America (SA, 80°W, 40°W); 3: Atlantic Ocean (AT, 40°W, 15°W); 4: Africa (AF, 15°W, 50°E); 5: Indian Ocean (IO, 50°E, 100°E); 6: Tropical West Pacific (TWP, 100°E, 180°); all these regions are with the same latitude range: 60°S - 60°N. (b) Example of the entry fraction in box region x = 1, $\theta$ > 400 K. The entered box region is vertically divided into 7 layers, from $\theta$ < 350 K, 350 K < $\theta$ < 360 K, ... to $\theta$ > 400 K, and horizontally divided into 6 regions shown in (a). The definition of the regions is adapted from Sun et al. (2023a).

stratosphere. Since the 380 K level is sufficiently high to facilitate the dominant vertical motion in the TTL (Gettelman and de Forster, 2002; Bergman et al., 2012), the air parcels at levels above 390 K are likely to enter the stratosphere as part of the BDC above the TWP. For higher levels in December, the air masses that ascend higher than the 400 K level account for 29% of the overall trajectory number, with 16% remaining over the TWP and 12% over the Indian Ocean. The dominance of easterly winds in both months corresponds to the observed Quasi-Biennial Oscillation (QBO) phase during December 2018 and August 2022 (Diallo et al., 2018). Easterlies in the lower stratosphere favor upward transport by reducing mixing with mid-latitude air, enhancing tropical stratospheric entry. In contrast, in August the AEF is concentrated at lower levels ($\theta$ < 360 K) compared to December. In 10 days, 66% of the trajectory points descend to levels below 360 K in August. The horizontal distribution in August is similar to that in December, corresponding with easterly winds in the lower stratosphere (Fig. 6) with most of the air masses moving towards the Indian Ocean and Africa.

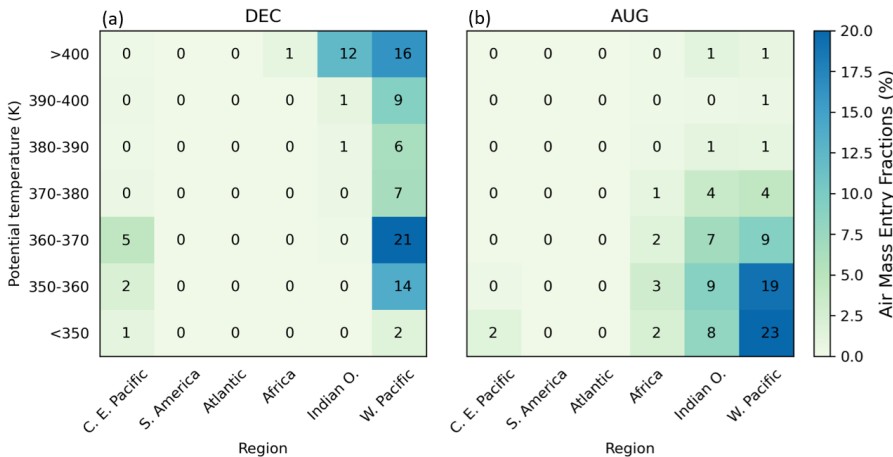

**Figure 8.** The air mass entry fractions (AEF) in (a) December 2018 and (b) August 2022. Calculations are based on the 10-d trajectory analysis, following the same approach as applied for Fig. 6.

## 3.5 Transport Pathways over the TWP

Understanding the sources and transport pathways of air masses over the TWP is crucial for unraveling the origins of atmospheric constituents of the global stratosphere (Rex et al., 2014). The inter-hemispheric mixing controls the origins of air

masses from the northern or southern hemisphere in the tropics. The seasonal movement of the Chemical Equator (CE) (Hamilton et al., 2008; Sun et al., 2023a) marks the boundary where air from both hemispheres converges and mixes, distinguishing air mass origins and influencing trace gas distributions. This dynamic separation reflects differences in anthropogenic and natural emissions between NH and SH hemispheres, characterizing the tropical atmospheric composition. Figure 9 summarizes the results of our study in a schematic diagram of transport pathways, considering the seasonal shift of the temperature structure

and inter-hemispheric mixing over the TWP. The approximate CPT in the tropics is located at 17 km within the TTL. The equilibrium between convective and radiative processes in the TTL controls air transport pathways (Randel and Jensen, 2013). The temperature in the TTL undergoes a seasonal cycle (see Fig. 1), which drives the seasonal shift of vertical transport. The spatial and temporal variations in the TTL associated with the dynamical circulation affect the transport pathways over the TWP, as shown by the arrows in Fig. 9a. Different cloud types, such as convection cells, thin cirrus, and SVC, are also depicted.

It should be noted that the monthly variations and gaps in the measurement periods of ComCAL can introduce uncertainties in our results. In August 2022 an intensive observational campaign including a particularly high number of lidar measurement hours was conducted at the PAO during ACCLIP. Even though the measurement time in December and August is very different, the percentage occurrence (PO) of cirrus clouds with 50.6% in December and 47.7% in August is within the acceptable difference between the two seasons (see Table. 1). This suggests that differences in measurement time do not necessarily

introduce larger uncertainties (Sun et al., 2024b). The observations in these two months are comparable.

Typical descending air masses in tropical regions after detrainment are illustrated by a downward pathway in both seasons in Fig. 9a and b. In the absence of convection vertical transport in the free troposphere and lower TTL is radiatively controlled and characterized by large-scale downwelling processes (Fueglistaler et al., 2009; Randel and Jensen, 2013). The forward trajectory analysis in Fig. 6e shows that in both seasons, the descending trajectories are primarily located at a level below 360

K, corresponding to the equilibrium level (EL) or the neutral buoyancy level (LNB) (Takahashi et al., 2017; Bergman et al., 2012). At this level, the main outflow from convective clouds occurs, and anvil clouds form. The prevalence of thicker cirrus cloud layers with higher optical depths compared to SVCs is typically detected at this level. This has been investigated and verified by ground-based (Sun et al., 2024b; Cairo et al., 2021; Pandit et al., 2015) and satellite observations (Takahashi et al., 2017).

The overshooting top is a rare but important pathway for tropospheric air entering the stratosphere, as shown in Fig. 9. During intense convection, such as severe thunderstorms, strong updrafts can propel warm, moist air above the CPT, forming a dome-shaped overshooting top that intrudes into the stratosphere (Wu et al., 2023; Fueglistaler et al., 2009). Although cold point overshooting tops account for only 1%-2% of all cirrus clouds above the CPT (Nugent and Bretherton, 2023), they play a key role in stratosphere-troposphere exchange (STE) by injecting air directly into the lower stratosphere.

The impact of overshooting convection on stratospheric humidity depends on the synoptic-scale tropopause temperature (Khaykin et al., 2022). As Fig. 9 shows, the overshooting top bypasses the CPT towards the warmer hemisphere. When the tropopause is cold, strong ice scavenging removes moisture, leading to dehydration of air parcels and cloud formation before they reach the stratosphere, as inferred from our lidar observations of cirrus clouds and relative humidity profiles in December. Conversely, when the tropopause is relatively warm, convective overshooting can bypass the cold trap and inject ice-rich air,

leading to stratospheric hydration referring to the August case we measured. This process influences the radiative balance and cirrus cloud formation. However, our measurements do not directly observe deep convection using ComCAL, as these clouds are too optically thick for backscatter detection. Instead, we infer the presence and effects of deep convection from seasonal comparisons of upper-level cirrus clouds combined with trajectory analyses. These cirrus clouds likely represent cloud detrainment after deep convection or dehydrated ice crystals formed from water vapor entering the TTL (Sun et al.,

2024b). Furthermore, overshooting convection rapidly transports short-lived species (e.g. $NO_x$, CO, and hydrocarbons) into the lower stratosphere, bypassing tropospheric oxidation processes, particularly over the TWP, where the oxidizing capacity is low. These chemical injections impact ozone chemistry and the stratospheric oxidation capacity, further altering the stratospheric composition.

## 4 Discussion

Our analysis gives evidence of the uplift of tropospheric air to the stratosphere via the cold trap during NH winter, which is consistent with previous studies (Fueglistaler et al., 2004; Schoeberl et al., 2019). The detrained air from convective clouds is transported to the cold trap regions, where the extremely cold and dry conditions favor the dehydration process of the air masses, which ultimately ascend to the stratosphere (Fig. 3 and Fig. 6a and c). The SVC and ETTCi are usually detected near

the CPT, providing insights into this dehydration process (Immler et al., 2007; Sun et al., 2024b). In contrast, air masses in
summer descend after reaching the CPT and do not reach the stratosphere, even though there are very similar cirrus cloud layer
cases measured by the lidar (Fig. 2a and Fig. 4a). This is consistent with previous studies, e.g., Fueglistaler et al. (2004); Krüger
et al. (2008); Bergman et al. (2012); Schoeberl et al. (2019), on the efficiency of vertical transport in the TTL, recognizing the
TWP as the dominant source region of stratospheric air in NH winter. The significantly stronger uplift during NH winter,
compared to NH summer, aligns with a stronger upwelling in NH winter due to the BDC, facilitating further upward transport.
The important role of tropical convection places geographical constraints on the source regions of stratospheric air, including
the TWP (Bergman et al., 2012; Schoeberl et al., 2019), the ASM anticyclone region (Honomichl and Pan, 2020a; Yan et al.,
2019; Pan et al., 2022, 2024; Vogel et al., 2019), the WPM region (Sun et al., 2023a; Savin et al., 2023) and specific land areas
(Nugent and Bretherton, 2023; Wu et al., 2023). Trade winds converge from both hemispheres, resulting in intense convective
activities and the formation of convective clouds. Convection here is further modulated intraseasonally by the Madden-Julian
Oscillation (MJO) and equatorial Kelvin waves, influencing atmospheric stability and vertical moisture transport. In addition,
ENSO phases, such as the weak El Niño in December 2018 and La Niña in August 2022, contribute to variability in tropical
convection patterns, which can impact the dynamical processes discussed here. Enhanced convection during the active MJO
phase and upward motion associated with Kelvin waves promote cirrus cloud formation in the upper troposphere by providing
the necessary cooling and moisture conditions (Kiladis et al., 2009; Kikuchi and Takayabu, 2004). The thin cirrus layers during
August measured above Palau can also be related to this convection-induced process. Furthermore, dynamics in the UTLS are
tied to the QBO (Diallo et al., 2018). As mentioned in Sect.3.4, the QBO is in the easterly phase in December 2018 and August
2022, which is consistent with the trajectory and AEF results. Recent research suggests that during the easterly phase of the
QBO, tropical upwelling within the BDC is strengthened (e.g. Rao et al. (2019)). However, the exact mechanisms and impacts
are still areas of active research, and further studies are needed to fully understand STE in connection with the QBO (Diallo
et al., 2018).

The origin of tropospheric air masses above the TWP is mainly controlled by the seasonal north-south movement of the
CE/ITCZ (Sun et al., 2023a; Müller et al., 2024a, b). During NH winter, the dominant air mass origins are within the NH (red
arrows and circles in Fig. 9), specifically Southeast Asia, transporting high concentrations of species related to human activities,
such as $O_3$ and CO to the TWP. This is characterized by measurements from the FTIR and ozone sonde data from previous
studies (Müller et al., 2024a, b; Sun, 2024a). The air masses from Southeast Asia are transported vertically through convective
towers into the TTL. There are low concentrations of OH and $O_3$, and thus a low oxidizing capacity, in the troposphere above
the TWP (Rex et al., 2014). In contrast, during summer, the CE is located north of Palau, resulting in clean air blowing from
the SH to Palau (Fig. 9b) and with the trade winds from the East, the west Pacific. Given the low frequency (1%-2% of all
tropical stratospheric cirrus (Nugent and Bretherton, 2023)) of cold point overshooting tops as a source of stratospheric air,
an upward pathway through the cold trap during winter is highly likely to transport tropospheric species into the stratosphere.
This scheme of stratospheric pathway over the TWP in winter significantly impacts the water budget and ozone chemistry in
the stratosphere and, consequently, on the global climate (Li et al., 2020; Villamayor et al., 2023). In the UTLS, the influence
of the ASM anticyclone on air mass transport towards the TWP becomes relevant, which potentially transports polluted air

masses into the local UTLS (Pan et al., 2022). The transport of pollutants in winter simultaneously alters the oxidizing capacity compared to the clean air scenario in TWP (Nicely et al., 2016). Further quantification of short-lived species through direct measurements, such as aircraft and satellite observations over the TWP in the cold trap in both seasons, is needed to validate this process.

The ASM region as an important entry point of air masses into the stratosphere in NH summer has been highlighted by several other studies and is still an active research field (Randel et al., 2010; Pan et al., 2022, 2024). The PAO measurements add to this discussion by their location at a transit point of air masses coming from the ASM and potentially entering the stratosphere elsewhere. Our results show a pathway of some cirrus cloud air masses from the ASM in August 2022 during the ACCLIP campaign. Honomichl and Pan (2020a), using ERA-Interim data from 1979–2017, analyzed transport pathways from the ASM via the western Pacific anticyclone to the western Pacific. Similar results from other studies (e.g., Randel and Park (2019); Pan et al. (2016)) highlight potential air mass transport routes toward the TWP. It highlights the TWP, South East Asia, and the Indian Ocean as stratospheric entry points of these air masses during the NH summer. While most of the cirrus cloud air masses observed above Palau do not further ascend into the stratosphere, given the weaker cold trap, there is a minor fraction of 14 % rising above 370 K potential temperature over the Maritime Continent and Indian Ocean (Fig. 8b), in compliance with previous studies. However, analyzing the ACCLIP aircraft measurements, Pan et al. (2024) report air mass tropopause crossing locations further north than our trajectory analysis suggests. This implies that the air masses observed by lidar above Palau were not sampled by the aircraft, which will need to be assessed further.

## 5 Conclusions

This study investigated the seasonal variability and transport pathways of air masses over the tropical western Pacific (TWP), focusing on the role of the cold trap in atmospheric dynamics and composition. Our findings show significant seasonal differences, with the TWP cold trap exhibiting more extreme temperatures and extent and greater dehydration of air masses during winter compared to summer.

As the December case study shows, the cold trap in the TWP is characterized by extremely low tropopause temperatures at higher altitudes. Under these conditions, air masses, particularly those associated with subvisible cirrus clouds (SVC) and extremely thin tropical cirrus (ETTCi) at altitudes near 18 km, undergo significant dehydration. During this process, high relative humidity with respective to ice (RHi) is also observed at the altitude of the cloud layer. Latent heat can be released because of the condensation of water vapor, which occurs as a result of ascent and extremely low temperatures due to the large-scale circulation in this region. In contrast, the cold trap is weaker during the NH summer, as shown by observations in August 2022, where CPT temperatures were higher around 190 K and the air was subsaturated. As a result, air masses showed a tendency to descend rather than ascend.

By defining the air mass entry fraction, we quantify the fraction of stratospheric air masses over different regions during different months. The results show that 46% of the December air masses ascend to temperature levels above 380 K, contributing to the stratospheric pathway. In August, only 5% of the air masses rise above the 380 K level above the TWP via the Asian

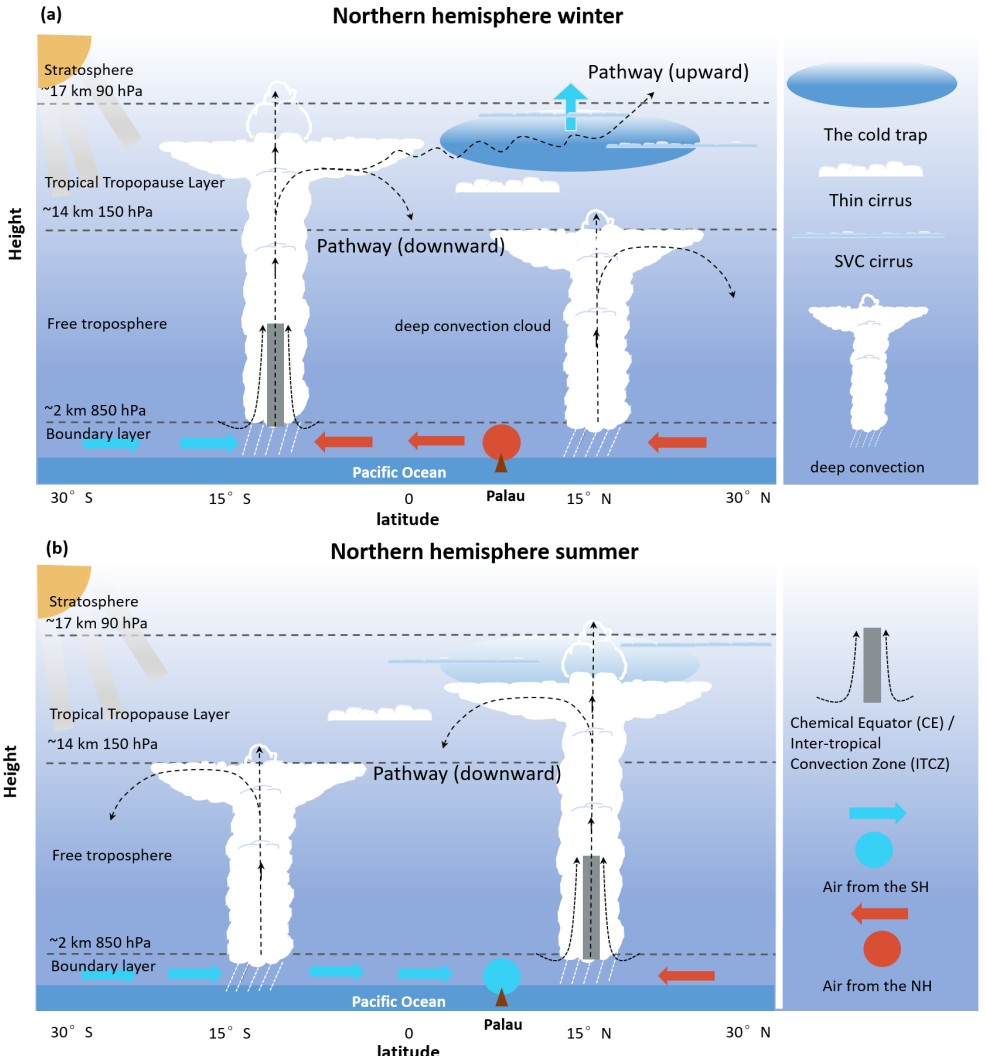

**Figure 9.** Schematic of transport pathways over the Tropical Western Pacific (TWP) related to seasonal shifts in circulation: the Intertropical Convergence Zone (ITCZ) or Chemical Equator (CE) and their effects on air source changes during (a) Northern Hemisphere (NH) winter and (b) NH summer. Seasonal ITCZ/CE shifts influence the inter-hemispheric mixing process, with red representing NH air and blue representing Southern Hemisphere (SH) air. The tropical site Palau (7.3°N, 134.5°E) is marked at the base of the figure. Air converges and rises in convective clouds, forming anvils before detaching from the main tower. After detrainment, three pathways occur: (1) Upward – air ascends through the cold trap, dehydrating under the formation of thin cirrus before reaching the stratosphere (NH winter); (2) Downward – air descends into the free troposphere or tropical tropopause layer (dominant in both seasons); and (3) during strong convection, some air may overshoot directly into the stratosphere. These pathways enable boundary layer and lower troposphere air masses to differ in composition and affect the stratosphere.

summer monsoon anticyclone pathway. The significantly higher AEF during NH winter also aligns with a strengthened BDC, which facilitates further upward transport.

This study highlights three pathways influencing air transport into the stratosphere: upward transport, predominant during NH winter, facilitating the dehydration and ascent of air masses through the cold trap into the stratosphere; downward transport, characterized by descending air masses, particularly during NH summer; and cold point overshooting tops, a rare (1%-2% of all tropical stratospheric cirrus) but momentous process where strong convective activities force air masses directly into the stratosphere.

The lidar observations in Palau and trajectory modelling provides further evidence of tropospheric air slowly ascending into the stratosphere above the TWP during NH winters. To account for the uncertainty inherent in trajectory calculations, we compared trajectory results from two Lagrangian transport models, ATLAS and HYSPLIT, confirming that both models consistently show the same seasonal transport patterns, with discrepancies primarily arising from differences in vertical coordinate. While the weaker cold trap in NH summer suppresses this pathway, our analysis captured the influence of the ASM and potentially related stratospheric entry of air masses above the Maritime Continent and central Indian Ocean. Our results highlight the critical role of the cold trap, where air masses are transported quasi-horizontally into the lower stratosphere, with possible implications for global stratospheric chemistry and water budgets. Direct observations within the cold trap, particularly of trace gases and cirrus clouds, remain limited and need to be further investigated. Observations are ongoing at the PAO, and there will be an increase in coinciding balloon-borne water vapor and aerosol measurements in the TTL. These efforts will address the observation and data gaps in the TWP. Future research focusing on these direct observations will be critical to validating these processes and improving our understanding of their implications for global climate dynamics.

## Appendix A: Comparison between HYSPLIT and ATLAS Trajectory Simulation

Diabatic heating plays a critical role in the UTLS. In the HYSPLIT model, vertical velocity is calculated using reanalysis data of vertical velocities derived from mass balance, known as kinematic vertical velocity. In the other approach, which accounts for the stratospheric radiation budget, the vertical velocity is driven by the heating rate. To investigate the sensitivity of trajectory pathways calculated with HYSPLIT setup, we use the additional, we use the additional trajectory module of the Lagrangian chemistry and transport model ATLAS (Alfred-Wegener-Institute LAgrangian Chemistry/Transport System) (Wohltmann and Rex, 2009; Wohltmann et al., 2010), which uses both vertical wind and heating rates in a hybrid approach to derive the vertical velocity.

With the ATLAS transport module trajectories are calculated backwards and forwards in time for 20 days using ERA5 reanalysis meteorological fields and diabatic heating rates with a horizontal resolution of 1.125° × 1.125° and a temporal resolution of 3 hours. This setup implicitly includes convective effects due to ERA5 convection parametrization, with a hybrid vertical coordinate that gradually changes from kinematic to fully diabatic above 100 hPa (Konopka et al., 2007). Trajectory starting points within a cirrus cloud layer are vertically spaced at 100 m intervals in geometric height and converted to pressure levels using radiosonde observations from the PWS, taken closest in time to the cirrus cloud observation. All trajectories within

a cirrus cloud are initiated simultaneously every hour during the time of the lidar observation and with an output time step of 1 h. For a detected cloud layer case, the number of trajectories is thus the cloud thickness divided by 100 m times the durations of the cloud observation in hour. In the scope of this study, the full inter-comparison between the two models is not the focus. The examples of case study simulations are given here to show the similar relative distribution of trajectories over the TWP, which is the focus of this study. In this sense, differences in reanalysis data inputs are neglected when comparing the sensitivity
of trajectory calculations through case studies.

Here, we compare the 20-day trajectory results from HYSPLIT and ATLAS for two case studies. For clarity, only 14 UTC for the winter case and 12 UTC for the summer case are shown; additional results are available in the supplement Fig. S4–11. In the December case, both models show a similar ascent of air masses before and after arrival in Palau (Fig. A1 and Fig. A2). Backward trajectories exhibit lower potential temperatures than forward trajectories, highlighting upwelling processes over
Palau. However, the forward trajectory pathways of the upper-level cloud differ significantly between models. The ATLAS model shows more westerly trajectories extending to northeast Africa, while HYSPLIT trajectories reach only the Indian Peninsula. HYSPLIT forward trajectories display a circling pattern within the TWP cold trap, while ATLAS primarily shows eastward movement, with only one circling trajectory at 14 UTC (Fig. A2b and Fig. A1b).

At lower levels, trajectory results are more consistent between models (Fig. A1a, c, and d / Fig. A2a, c, and d), suggesting
that differences in reanalysis data are less impactful than those in vertical velocity approaches. For upper layers, ATLAS's hybrid vertical coordinate uses diabatic heating rates above 17 km, while HYSPLIT employs kinematic velocities. These differences likely explain the observed discrepancies. As noted by Schoeberl and Dessler (2011), kinematic methods offer a higher probability of encountering low temperatures than diabatic methods, this may lead to overestimated geopotential heights in HYSPLIT trajectories. This effect is evident in December when lower temperatures of the cold trap intensify temperature
gradients. HYSPLIT trajectories therefore frequently exhibit circling patterns within the colder cold trap, whereas ATLAS shows only a single case of such a pattern. In August, due to the warmer cold trap and weaker temperature gradients, differences between the two models are less pronounced.

Cross-sections for the December case at 14 UTC (Fig. A5) further illustrate these differences. The trajectories in the ATLAS model do not rise above a pressure of about 70 hPa as in HYSPLIT (Fig. A5a and b), but the potential temperatures in the cold
trap for both models exceed 390 K. Additionally, some HYSPLIT forward trajectories show an abrupt, strong uplift above 18 km close to 20 days (Fig. A5b and d), while is not the case for ATLAS trajectories (Fig. A5a and c). This strong uplift is likely an artificial effect in kinematic vertical velocity calculations, as previously noted in trajectory studies (Setc. 3.2). To further ensure robustness and to avoid such unrealistic uplifts, we specifically selected 10-day trajectories for the monthly analysis in Sect. 3.3, and the calculation of AEF in Sect. 3.4, supporting the conclusions drawn in Sect. 5. However, since this feature
occurs only at the very end of the 20-day trajectory, it does not affect our conclusions regarding seasonal ascent in winter and descent in summer, which remain consistent based on the combined forward and backward trajectory analysis. Both models provide evidence of the predominance of ascending air masses in Case 1 (the winter case) and descending air mass in Case 2 (the summer case) over Palau. A thorough model inter-comparison accounting for differences in both vertical velocity and

**Case 1: 13 December 2018 at 14 UTC**

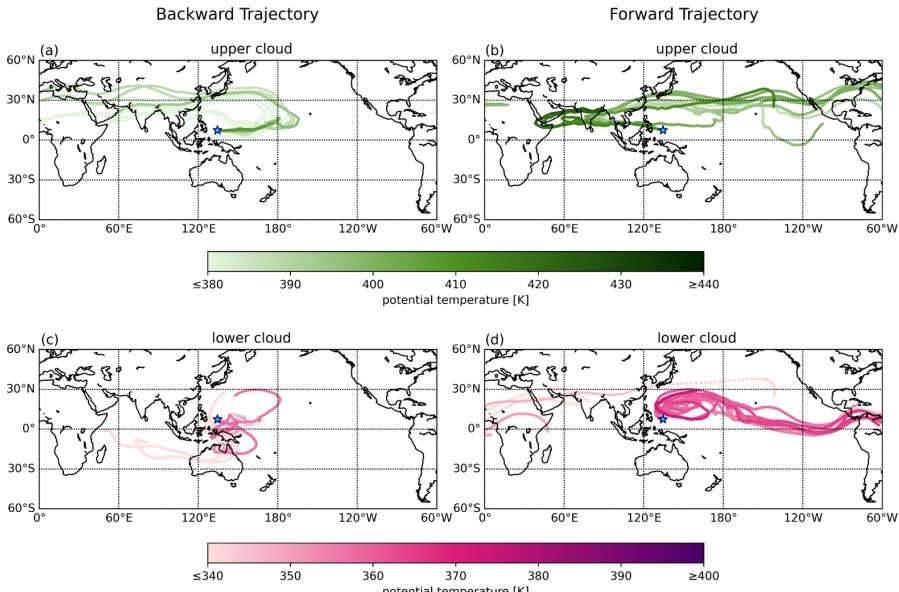

**Figure A1.** 20-d trajectories of the upper cloud (a and b, upper row), and the lower cloud (c and d, lower row) at 14 UTC for the winter case (13 December) by ATLAS. The left column (a and c) shows backward trajectories and the right column (b and d) shows forward trajectories. The upper cloud altitude is 17.5–18.1 km, and the lower cloud altitude is 14.6–16.1 km.

reanalysis datasets would further clarify remaining uncertainties and add new insights to the discussion of transport studies in
the TTL (Honomichl and Pan, 2020b).

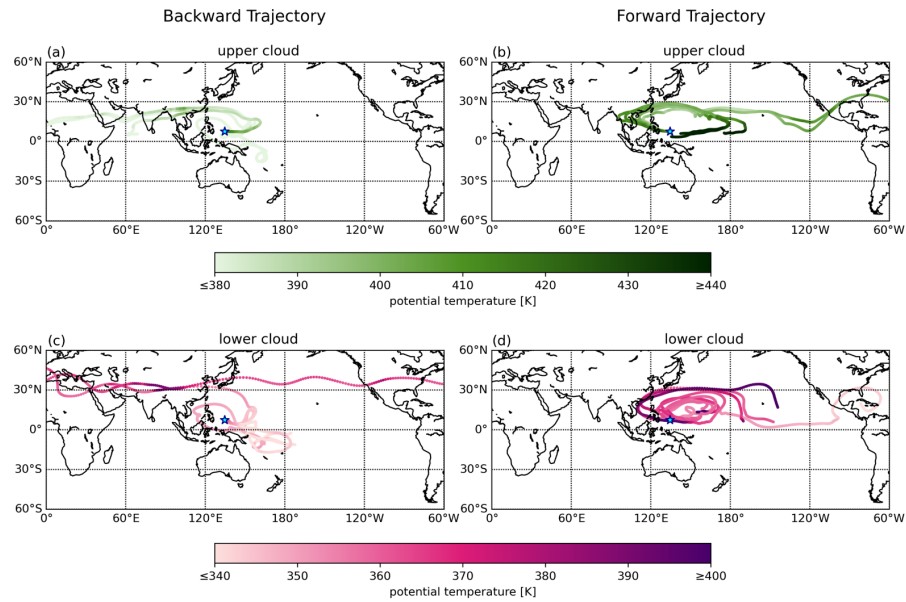

**Figure A2.** As Fig. A1 but for HYSPLIT.

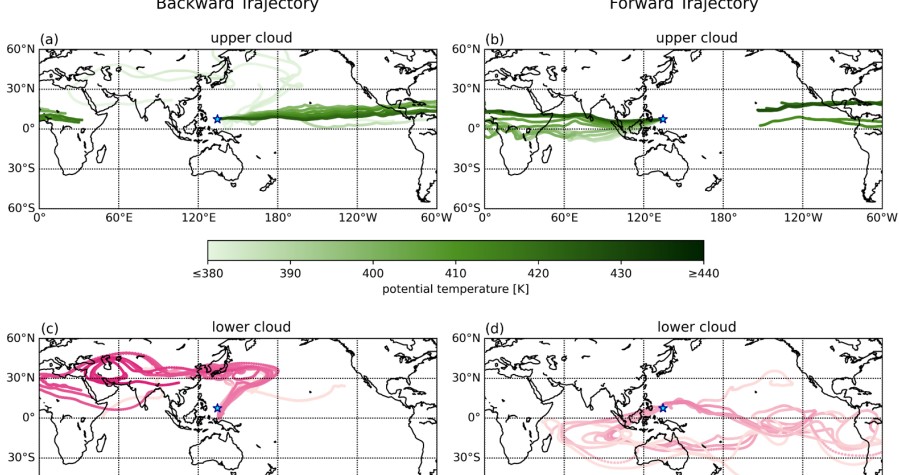

**Figure A3.** As Fig.A1 but for Case 2 (summer) at 12 UTC by ATLAS. The upper cloud altitude is 17.1–17.7 km, and the lower cloud altitude is 13.4–14.4 km.

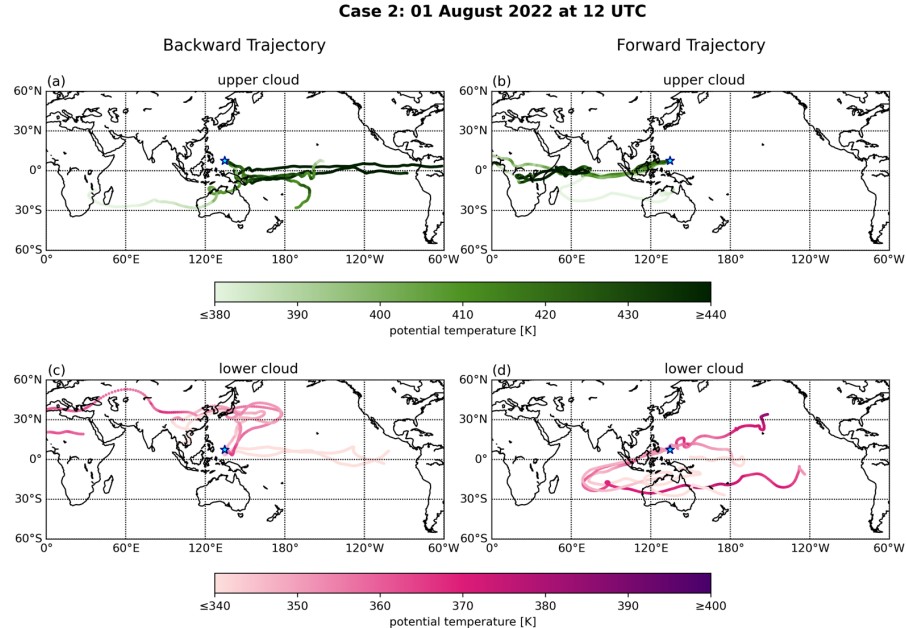

**Figure A4.** As Fig. A3 but for HYSPLIT.

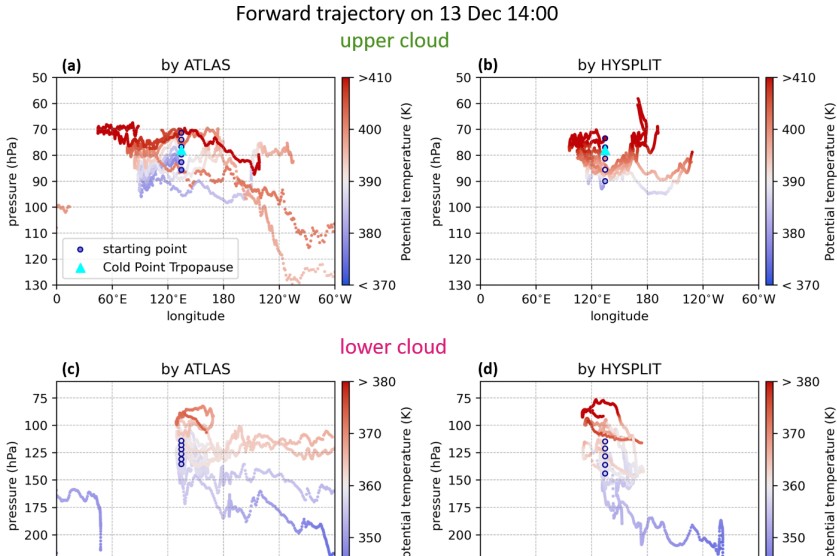

**Figure A5.** Longitude-pressure cross-section of the 20-d forward trajectories of the upper cloud (a–b) and the lower cloud (c–d) at 14 UTC for the winter case by ATLAS (left column) and HYSPLIT (right column), color-coded by potential temperature. The upper cloud altitude is 17.5–18.1 km, and the lower cloud altitude is 14.6–16.1 km. The cold point from the radiosonde (see Fig. 2 d) is marked by the cyan triangle (a–b). The starting points of the trajectories associated with cirrus cloud measurements (see Fig. 2 a) are marked by circles in each plot.

*Acknowledgements.* The authors want to thank Patrick Tellei, President of the Palau Community College, for the provision of space for the laboratory containers in the college; German Honorary Consul Thomas Schubert, for overall support; and various people and institutions for operations at the PAO: Jürgen "Egon" Graeser (AWI), Ingo Beninga (Impres GmbH), Wilfried Ruhe (Impres GmbH), Winfried Markert (Uni Bremen), and Sharon Patris (Coral Reef Research Foundation, Palau). We want to further thank Laura Pan (NCAR) and the ACCLIP science team for the discussions and the overall successful campaign. This work has been supported by the BMBF (German Ministry of Research and Education) in the project ROMIC-II subproject TroStra (01LG1904A) and Central Research Development Fund (CRDF) of the University of Bremen, ZF 04 (Nr. 0100295604).

*Code and data availability.* The dataset of the ComCAL measurements related to this publication has been partly made available (during the Asian Summer Monsoon Chemical & Climate Impact Project (ACCLIP) campaign, August 2022) in the UCAR/NCAR-Earth Observing Laboratory: https://doi.org/10.26023/SD3W-WKBH-5X13, Sun et al. (2023b). Other data products of ComCAL are available upon request from the corresponding author. The Palau weather station data (station code: PTRO 91408) was accessed via the upper-air sounding database provided by the University of Wyoming, https://weather.uwyo.edu/upperair/sounding.html, last accessed on 13 April 2024. The monthly temperature and pressure data were accessed via the Copernicus Climate Data Store, https://doi.org/10.24381/cds.6860a573 (Hersbach et al., 2023), last accessed on 10 June 2024. The tropopause data are from the Reanalysis Tropopause Data Repository, https://doi.org/10.26165/

JUELICH-DATA/UBNGI2, (Hoffmann and Spang, 2021), last accessed on 10 June 2024. The meteorology data for the HYSPLIT run are available at https://www.ready.noaa.gov/data/archives/gdas1/, last accessed on 12 November 2024.

     The HYSPLIT model code used in this analysis is downloaded from https://www.ready.noaa.gov/HYSPLIT.php, last accessed on 10 June 2024. The ATLAS model code used in this study is available upon request from the corresponding author.

*Author contributions.*  XS and MP designed the study. XS carried out the model simulations and the data analysis. XS and CR conducted
and validated the lidar measurements. KM coordinates the PAO instrument operations and contributes to the design of the method. DJ, KM, and TR performed the trajectory model simulation. XS wrote the manuscript, with contributions from all coauthors. All authors helped to discuss the results and comment on the manuscript.

*Competing interests.*  The authors have declared that they don't have any competing interests.

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
