# Peer review of "Evidence of Tropospheric Uplift into the Stratosphere via the Tropical Western Pacific Cold Trap"

_EGUsphere, 2024_

## Author Comment (AC1)

Response to Comments of Reviewer 1

*The authors thank all reviewers for their constructive comments and suggestions, which have helped us to improve the quality of this paper both in sciences and writing. All comments are carefully considered and responded to. The response in italic letters follows each comment.*

This manuscript presents a well-structured and scientifically significant study on air mass transport pathways over the Tropical Western Pacific (TWP), with a particular focus on the role of the cold trap in the upper troposphere and lower stratosphere (UTLS). Using lidar and balloon-borne observations from Koror, Palau, combined with trajectory simulations, the authors provide valuable insights into seasonal variations in air mass ascent and dehydration processes.

The study is particularly relevant as it enhances our understanding of how air masses transition from the troposphere to the stratosphere, a process fundamental for stratospheric water vapor balance and global climate dynamics. The authors effectively demonstrate the stark contrast between winter (December 2018) and summer (August 2022) conditions, showing that upward transport through the cold trap occurs primarily in winter, while summer air masses tend to descend. The inclusion of quantitative metrics, such as the fraction of air masses reaching above 380 K potential temperature, strengthens the findings and makes a compelling case for the seasonal dependency of stratospheric entry pathways.

Overall, this is an important and well-executed study that deserves publication. I only have a few minor suggestions regarding clarity, phrasing, and data presentation that could further improve the manuscript. These are detailed below:

(151). The use of two case studies (December 2018 and August 2022) allows for a seasonal comparison, adding depth and clarity to the analysis. However, in both case studies there is no description of the meteorological conditions that would have provided additional information about the local context in which the measurements were taken.

*Response:*

*Thank you for your comment. We agree that the meteorological context is important for interpreting the case studies. In our analysis, we use radiosonde data to provide temperature and humidity profiles that are directly matched to the observation time and location. These profiles capture the local atmospheric conditions relevant to our study, including tropopause temperature and moisture distribution, which are key factors influencing cirrus cloud formation and stratospheric transport. Given that the radiosonde measurements offer high temporal and spatial relevance to our case studies, we chose not to include additional meteorological field data. However, we will clarify this in the text to ensure that the role of the radiosonde data in providing meteorological context is explicitly stated.*

*We rewrote the following sentence:*

*"We analyzed two typical cases of cirrus cloud layers measured in December and August, combining meteorological conditions including temperature and RH profiles obtained from the radiosonde, measurements of cirrus clouds in the tropical tropopause layer (TTL), and trajectory simulation results."*

(176). There is no information here on where precisely the backtrajectories are initiated. Midcloud maybe? One trajectory per cloud? Clusters? You may consider to shift here (if appropriate) lines 417-420 from appendix A.

*Response:*

*We initiated the HYSPLIT trajectory from the cirrus cloud layers, specifically 5 trajectories between the cloud top and base height in each layer. The trajectories are started every hour corresponding to measurement time from ComCAL lidar observations. For ATLAS, trajectory starting points are spaced at 100 m interval within each cloud layer. The releasing time is the same as HYSPLIT with 1 h interval. The info are given in L115-122 and as you've mentioned L417-420.*

*We add some details about the trajectory setup and add a reference note here to the method section, in L178:*

*"To further analyze the STE pathways for different cloud types on this day, we present 20-day backward and forward trajectories from the upper and lower cloud layers in Fig. 3. Ascent is shown in red, while descent is shown in blue. Additionally, for long-lasting cirrus clouds, one back and one forward trajectory is initiated every hour. So within each cloud layer, multiple trajectories are initiated to capture variations. The trajectory setup details are described in Sect. 2.3 and Appendix A."*

Fig2. It would be nice that panel 2a, 2c and 2d could share the same vertical axes. This can be done by removing panel 2 b. The latter is cited in the text but not further discussed so it might be sufficient to state in the text the amount, variability and trends (if any) of the COD, and skip the figure. By the way, in the figure 2b there appear to be a lack of COD data between 11.5 and 12.5 and 13.15-14.15. Why? Moreover, in fig2a the colors are coded in a.u. Why? Can you plot explicitly the BR?
Response:

*We appreciate the reviewer's suggestions regarding Figure 2. However, we believe that keeping panel 2b is important for the following reasons:*

- *Scientific Context: Panel 2b provides essential information for cirrus cloud classification based on COD values. As COD is critical for distinguishing different cirrus types—ETTCi (extremely thin cirrus, COD < 0.005), SVC (subvisible cloud, COD < 0.03), thin cirrus (COD < 0.3), and thick cirrus (COD > 0.3)—this panel allows readers to understand the variability and distribution of cloud types directly from the figure. We updated the plot (Panel 2b) and marked the different cirrus types based on COD.*

[Figure]

*Figure 1 (Fig.2b and Fig. 4b in the manuscript). The backscatter ratio (BSR) at 532 nm as a function of time and altitude. Thin cirrus (COD < 0.3), SVC (Sub-visible cirrus, COD < 0.03) and ETTCi (extremely thin cirrus, COD < 0.005) are marked.*

- *Trajectory and Seasonal Analysis: Panel 2b also supports the discussion of how different cirrus types, identified through their COD values, exhibit distinct seasonal patterns when combined with trajectory analysis. Removing this panel would weaken the connection between COD-based classification and the cloud variability discussed later, for example in Sect. 4 L295 "Different cloud types, such as SVC…".*

*Based on the reviewer's suggestions, we have added more descriptions and discussions of cloud COD in the text, clarifying its importance for cloud classification and seasonal variations.*

*Regarding the reviewer's specific questions:*

*Missing COD Data in Figure 2b: The absence of COD data between 11.55–12.61 and 13.21–14.28 is because the lidar was turned off during these intervals, as confirmed by our measurement records. The apparent continuous coloring in the pcolormesh plot of BSR (panel a) results from interpolation between available time points. We masked these time periods in the revised manuscript to avoid confusion.*

*Color Coding and BSR in Figure 2a: we thank the reviewer for the question regarding the units of the backscatter ratio (BSR) in Figure 2a. BSR is a dimensionless quantity because it is defined as the ratio of the total backscatter coefficient $\beta_{total}(\lambda)$ to the molecular (Rayleigh) backscatter coefficient $\beta_{Ray}(\lambda)$ :*

$$BSR = \frac{\beta_{total}(\lambda)}{\beta_{Ray}(\lambda)} = \frac{\beta_{par}(\lambda) + \beta_{Ray}(\lambda)}{\beta_{Ray}(\lambda)} = 1 + \frac{\beta_{par}(\lambda)}{\beta_{Ray}(\lambda)} \tag{1}$$

*where $\beta_{par}(\lambda)$ is the backscatter coefficient of particles, and the total backscatter is simply the sum of the contribution of the individual components. When the BSR value is greater than 1, there is more backward scattering in the atmosphere in addition to Rayleigh scattering. This is the amount available for assessing the strength of atmospheric backward scattering, which in this study is cirrus cloud.*

*Since both $\beta_{total}(\lambda)$ and $\beta_{Ray}(\lambda)$ have the same units (typically $m^{-1}sr^{-1}$) their ratio is unitless. Therefore, BSR does not have a physical unit and is expressed in arbitrary units (a.u.) on the color scale, indicating relative intensity rather than an absolute physical measurement.*

Fig.3. : See the remarks on Fig. 2

*Please see the response to Fig. 3 above.*

(205-207). In this sentence, it almost seems as if dehydration is a cause for upwelling, and it could be rephrased to avoid giving this impression. The Brewer-Dobson Circulation provides the dominant large-scale upwelling mechanism in the TTL, and deep convection and cold trap dehydration regulate the water vapor content, influencing the efficiency of air entering the stratosphere. While it is true that latent heat release from cirrus formation enhances local ascent, the radiative effects of cirrus can either amplify or dampen vertical motion depending on cloud properties. Therefore, the role of latent heat release by condensing cirrus in TTL ascent can be complex and not straightforward.

*Response:*

*Thank you for your suggestion. We see how the wording might imply a causal relationship between dehydration and upwelling, which was not our intention. We will revise the sentence to clarify that the Brewer-Dobson Circulation (BDC) is the primary driver of large-scale upwelling in the TTL, while deep convection and cold trap dehydration regulate water vapor content, thereby influencing stratospheric entry efficiency. We also appreciate the note on the radiative effects of cirrus clouds, and we will refine the discussion to better capture the complexity of their role in TTL ascent.*

*We rewrote and added the following sentences in L207:*

*"The lower and sub-saturated RHi below the CPT suggests that the cold trap during this period is not cold enough to further dehydrate the air parcels passing through it. Cloud formation in the cold trap releases latent heat, which may influence local temperature and stability. However, the radiative effects of cirrus clouds can either amplify or suppress vertical motion, depending on the ice crystal shape and size (Spang et al., 2024). Specifically, ultrathin cirrus clouds composed of spherical ice crystals tend to warm the atmosphere, while aggregated or hexagonal ice crystals lead to cooling. These cloud microphysical effects interact with large-scale circulation, potentially influencing the vertical transport patterns observed in the trajectories."*

*We also added the following sentences in L240 to strengthen the connection between the BDC and our analysis:*

*"A comparison of observed SVC and ETTCi and the associated trajectories shows that clouds with similar optical depth nevertheless ascend in winter and descend in summer. This contrast aligns with the seasonal variability of the BDC, which is stronger in NH winter, supporting enhanced upwelling over the TWP due to an extremely cold trap. In contrast, during NH summer, the BDC is weaker over Palau, and there is no evident large-scale upward motion shown in trajectory results. This implies that seasonal differences in the strength of the BDC play a key role in modulating the transport pathways, rather than cloud microphysical processes alone. …"*

(229). The potential influence of Kelvin waves is a valuable addition to understanding the August case. Is there direct observational evidence, as temperature anomalies or reanalysis data that confirm the presence of such waves?

*Response:*

*We have not yet conducted a specific analysis using reanalysis data over Palau for this study. Here, we propose Kelvin waves as a possible explanation based on previous studies, e.g. Immler et al., 2008, that have demonstrated their connection with cirrus clouds using reanalysis data. We plan to investigate this relationship over Palau in greater detail soon, leveraging targeted reanalysis data and cloud observations.*

(243-261) The section nicely extends the previous case studies by performing trajectory analyses for all cirrus clouds in two seasons. The inclusion of potential temperature analysis strengthens the transport pathway discussion. However, there is no information on where the trajectories are initiated in case of geometrically thick clouds. Midcloud? Or if the geometrical thickness is large, more than a backtrajectory is used for the same cloud? And for long lasting cirrus observation, do you launch a trajectory every 3 hrs? The sentence at (248) "The trajectories are initialized at the observed time and altitude of cirrus clouds above Palau, consistent with the case study methodology." should be expanded to provide such information explicitly. This has an impact on the understanding of the AEF afterward.

*Response:*

*Thank you for your thoughtful comments. The details regarding the trajectory initialization are already described in the Methods section (Sect. 2.3). As noted there, in our HYSPLIT simulations, we divide the cirrus cloud layer into five levels, and in the ATLAS simulations, back trajectories are released every 100 m. Therefore, for geometrically thick clouds, multiple back trajectories are indeed used. Additionally, for long-lasting cirrus observations, we initiate one back or forward trajectory per hour. These settings are consistently applied to both the case study and the full-month analysis. We will expand the sentence at line (248) to explicitly clarify this information in the Results section.*

*We will refer to the method section (Sect. 2.3) at the beginning of this section to enhance the clarity.*

*This is the rewritten sentences:*

*"Similar as the case studies, the trajectories are initiated for the time and altitude of cirrus clouds measured above Palau. Within the cloud layers detected by lidar, multiple trajectories are used to capture variations. The trajectory setup details are described in Sect. 2.3 and Appendix A."*

(265) How are different box regions (1-6) defined? In terms of continental/maritime convection? Presence of monsoon? Please explain in further detail the reason for this boxing choice.

*Response:*

*Thank you for your question. We use the regional definitions (regions 1–6) following Sun et al. (2022), which extended the tropical-based definitions of Fueglistaler et al. (2004) to ±60° latitude. However,*

*since this study specifically focuses on tropical regions, we restrict our analysis to ±30° latitude, maintaining the original definitions and boundaries adapted from Sun et al. (2023).*

*We added the following sentences:*

*"The regional definitions (regions 1–6) used here follow Sun et al. (2023), adapted from Fueglistaler et al. (2004), but restricted to ±30° latitude to maintain a tropical focus."*

(267). What does N represent? Total trajectory points? Please state that explicitly.

*Yes, the total trajectory points. We rewrote this as following:*

*"where N (x,ϑ) represents the total number of trajectory points in the specified box region. Here, x ranges corresponding to different geographic regions, and ϑ represents potential temperature, ranging from ≤ 350 K to ≥ 370 K in 10 K intervals. For instance, N (1, ϑ > 400 K) represents the number of trajectory points in the central east Pacific region within the vertical layer above 400 K (as indicated by the dark green box in Fig. 7b. The AEF is then calculated as the fraction of trajectory points in each box region to the total number of trajectory points."*

(275) The text presents a strong seasonal contrast (46% vs. 5%) but does not explicitly explain why December favors stratospheric entry.  My suggestion: "The significantly higher AEF in December (46% reaching  380 K is consistent with stronger upwelling over the TWP during NH winter. This aligns with the seasonal phase of the Brewer-Dobson Circulation, which facilitates upward transport of TTL air into the stratosphere."

*Response:*

*Thank you for your insightful comment. We agree that the seasonal contrast in AEF suggests a stronger upwelling in December (NH winter). The suggested explanation aligns well with the seasonal phase of the Brewer-Dobson Circulation, which enhances the upward transport of TTL air into the stratosphere during NH winter. We have rewritten the text as you suggested to explicitly state this connection and clarify why December favors stratospheric entry.*

(279) The reference to the easterly upper-level winds and QBO connection need more clarity: The role of QBO-driven zonal wind patterns should be briefly explained.  My suggestion: "The dominance of easterly winds in both months corresponds to the observed QBO phase during December 2018 and August 2022 (Diallo et al., 2018). Easterlies in the lower stratosphere favor upward transport by reducing mixing with mid-latitude air, enhancing tropical stratospheric entry."

*Response:*

*Thank you for your suggestion. We agree that the connection between QBO-driven zonal wind patterns and stratospheric entry should be more clearly stated. The proposed explanation aligns well with our*

*findings, as the dominance of easterly winds during both months is consistent with the observed QBO phase in December 2018 and August 2022. We have revised the text here as you suggested to explicitly clarify that easterlies in the lower stratosphere reduce mixing with mid-latitude air, thereby enhancing the upward transport of tropical air into the stratosphere.*

(289). A better Explanation of the Chemical Equator (CE) is here needed. Add a brief sentence explaining what the CE represents and why it matters for tropical atmospheric composition.

*Thank you for your comment. We agree that a clearer explanation of the Chemical Equator (CE) would improve the readability of the text. We will add a brief sentence to define the CE and explain its significance in tropical atmospheric composition, particularly its role in distinguishing between air mass origins from NH or SH. We rewrote L289-291 as the following sentences:*

*"The inter-hemispheric mixing controls the origins of air masses from the northern or southern hemisphere in the tropics. The seasonal movement of the Chemical Equator (CE) (Hamilton et al., 2008, Sun et al., 2023) marks the boundary where air from both hemispheres converges and mixes, distinguishing air mass origins and influencing trace gas distributions. This dynamic separation reflects differences in anthropogenic and natural emissions between hemispheres, characterizing the tropical atmospheric composition. Figure 9 summarizes the results of our study in a schematic diagram of transport pathways, considering the seasonal shift of the temperature structure and inter-hemispheric mixing over the TWP. …"*

(330-337) Here consider to quote "Khaykin, S. M., Moyer, E., Krämer, M., Clouser, B., Bucci, S., Legras, B., Lykov, A., Afchine, A., Cairo, F., Formanyuk, I., Mitev, V., Matthey, R., Rolf, C., Singer, C. E., Spelten, N., Volkov, V., Yushkov, V., and Stroh, F.: Persistence of moist plumes from overshooting convection in the Asian monsoon anticyclone, Atmos. Chem. Phys., 22, 3169–3189, https://doi.org/10.5194/acp-22-3169-2022, 2022." and the dual role of overshooting convection, which may lead to hydration or dehydration depending on the synoptic-scale tropopause temperatures.

In general, this paragraph needs more clarity on the role of overshooting tops: First, explain why overshooting convection bypasses the cold trap and directly injects air into the stratosphere. Then, explain the impact of short-lived species.

*Response:*

*Thank you for your insightful comment. We agree that the dual role of overshooting convection in regulating stratospheric hydration and dehydration should be more clearly explained. We've revised the paragraph to explicitly describe how overshooting convection bypasses the cold trap, depending on the synoptic-scale tropopause temperature, as highlighted in Khaykin et al. (2022). We added the following sentences to L351 regarding the role of overshooting convection:*

*"The overshooting top is a rare but important pathway for tropospheric air entering the stratosphere, as shown in Fig. 9 in the white dashed line. During intense convection, such as severe thunderstorms, strong updrafts can propel warm, moist air above the CPT, forming a dome-shaped overshooting top that intrudes into the stratosphere (Wu et al., 2023; Fueglistaler et al., 2009). Although cold point overshooting tops account for only 1%-2% of all cirrus clouds above the CPT (Nugent et al., 2023), they play a key role in stratosphere-troposphere exchange (STE) by injecting air directly into the lower stratosphere.*

*The impact of overshooting convection on stratospheric humidity depends on the synoptic-scale tropopause temperature (Khaykin et al., 2022). As Fig. 9 shows, the overshooting top bypasses the CPT towards the warmer hemisphere. When the tropopause is cold, strong ice scavenging removes moisture, leading to dehydration of air parcels and cloud formation before they reach the stratosphere, as we measured in December. Conversely, when the tropopause is relatively warm, convective overshooting can bypass the cold trap and inject ice-rich air, leading to stratospheric hydration referring to the August case we measured. However, our measurements do not directly observe deep convection using ComCAL, as these clouds are too optically thick for backscatter detection. Instead, we infer the presence and effects of deep convection from seasonal comparisons of upper-level cirrus clouds combined with trajectory analyses. These cirrus clouds likely represent cloud detrainment after deep convection or dehydrated ice crystals formed from water vapor entering the TTL (Sun et al., 2024). This process influences the radiative balance and cirrus cloud formation. Additionally, overshooting convection rapidly transports short-lived species ($NO_x$, CO, and hydrocarbons) into the lower stratosphere, bypassing tropospheric oxidation processes, particularly over the TWP, where the oxidizing capacity is low. These chemical injections impact ozone chemistry and the stratospheric oxidation capacity, further altering the stratospheric composition."*

---

## Author Comment (AC2)

Response to Comments of Reviewer 2

*The authors thank all reviewers for their constructive comments and suggestions, which have helped us to improve the quality of this paper both in sciences and writing. All comments are carefully considered and responded to. The response in italic letters follows each comment.*

Review of "Evidence of Tropospheric Uplift into the Stratosphere via the Tropical Western Pacific Cold Trap"by Sun et al.

The authors use lidar and balloon measurement data, along with the trajectory model, to show the transport pathway over the tropical western Pacific cold trap. By comparing two typical cases, they find that the winter case shows the lowest temperature, coinciding with the RHi >150%. Nearly half of the air particles in the December cirrus cloud rise above 380 K after cloud formation. In contrast, the summer case shows a warmer tropopause and a non-supersaturated environment (RHi <100%). Only 3% of the air particles in the August rise above the 380 K level. The authors provide evidence of the uplift of tropospheric air to the stratosphere via the cold trap during NH winter. Analysis of such simulations is valuable and it is crucial to highlight the importance of the cold trap in driving air mass transport. I propose accepting the paper after the minor revisions listed below:

Line 27-29: The following ref. Such as Vogel et al., (2019) maybe useful.

Ref. Vogel, B., Müller, R., Günther, G., Spang, R., Hanumanthu, S., Li, D., Riese, M., and Stiller, G. P.: Lagrangian simulations of the transport of young air masses to the top of the Asian monsoon anticyclone and into the tropical pipe, Atmos. Chem. Phys., 19, 6007–6034, https://doi.org/10.5194/acp-19-6007-2019, 2019.

*Response:*

*Thank you for the suggestion. Vogel et al. (2019) provide valuable insights into the Lagrangian transport of young air masses into the Asian monsoon anticyclone and the tropical pipe, which aligns well with our discussion on air mass transport pathways. We've incorporated this reference to strengthen our stratosphere-troposphere exchange (STE) analysis and further support our findings.*

Line 30: ...in the lower atmosphere... to ...in the lower stratosphere...

*Corrected.*

For table 1: How should the mean cloud base or top height be considered for double cirrus layers in the UTLS region? Such as the cases on 13 December 2018 and 1 August 2022.

*Response:*

*Thank you for bringing up this important point. In cases with double cirrus layers, such as those on 13 December 2018 and 1 August 2022, we treat each layer as a distinct cloud and calculate the mean cloud base and top heights individually for each identified layer. Table 1 is intended to illustrate that the cloud properties observed during these two case studies closely match the monthly mean, thereby demonstrating their representativeness of typical cirrus layers over Palau. For more comprehensive*

*details regarding the calculation methods, cloud layer heights, temperatures, and the handling of multi-layer cloud cases, we kindly refer readers to our previous publication, Sun et al. (2024), which provides an extensive discussion on cloud properties across different seasons.*

Line 98: the mid-cloud temperature? Please give more information.

*We appreciate your suggestion for clarity. The mid-cloud temperature refers to the temperature at the vertical midpoint of each cloud layer, calculated by averaging the temperatures at the cloud top and cloud base. To improve clarity, we have revised line 98 as follows:*

*"The cloud base and top heights are determined for each cloud layer, and the mid-cloud temperature is defined as the temperature at the midpoint of each layer, calculated as the average of the temperatures at the cloud top and base heights. If more than one cloud layer is observed simultaneously, they are considered as separate cloud layers."*

*And we rewrote the paragraph regarding the cirrus cloud properties:*

*"These two months represent two distinct seasonal features of local cloud layers (Sun et al., 2024). The primary geometrical properties of the cirrus clouds observed over Palau are summarized in Table 1. The cloud base and top heights listed in Table 1 are determined individually for each cloud layer. The mid-cloud temperature is defined as the temperature at the midpoint of each cloud layer, calculated as the average of the cloud top and base heights. If more than one cloud layer is observed simultaneously, they are considered as separate cloud layers. Detailed discussions on quantifying cirrus cloud properties such as cloud layer heights, temperature, and the treatment of multi-layer clouds are given in Sun et al. (2024).*

*Table 1 demonstrates that the selected cases from December and August effectively represent typical cirrus cloud conditions observed during these two distinct seasons over Palau. The mid-cloud temperature in December is generally lower, and the cloud layers are higher compared to August, reflecting seasonal variations in meteorological conditions associated with the cold trap. These seasonal differences will be further described in Sect. 3.1, supporting reasons for selecting these two months for the following analysis."*

Line 136: Figure 1a and b... to Figures 1a and b...

*Corrected.*

Line 142: ...over the tropical region (±30°N)... to ...over the tropical region (30°S-30°N)...

*Corrected.*

Figure 1: why were reanalysis data from different periods used for Fig. 1a-b (1980-2019) and Fig. 1c-d (1992-2022)? The cold point temperature is usually not used in mid- and high-latitude regions.

*Response:*

*Thank you for your comment. The reanalysis data periods differ because Fig. 1a-b uses cold point temperature (CPT) data from Hoffmann and Spang (2022), which is only available up to 2019. However,*

*the reanalysis data in Fig. 1c-d is directly derived from ERA5 reanalysis, and since our study focuses on the period 2018-2022, we extended the ERA5 dataset to 2022 for consistency with our observations. We updated the reanalysis data for Fig. 1c-d 1980-2022 to keep these two datasets with the same starting time.*

*We acknowledge that CPT is generally not used in mid- and high-latitude regions. However, in Fig. 1a-b, we included data up to 60°S–60°N for completeness of visualization. In contrast, Fig. 1c-d does not include mid- and high-latitude regions in the averaging, ensuring the focus remains on the tropical regions relevant to our study.*

Line 263:...the relevant/dominant transport...?

*Corrected as "the dominant transport".*

Figure 8: ...on the 10-d trajectory analysis (compare 6). (compare 6)? more details

*Corrected as "following the same approach with Fig. 6."*

Line 387: In August, only 3% of the air masses... In the discussion and sect. 3.4, the value is 5%?

*Corrected here to 5%.*

For Figures A1-A4: The differences between the ATLAS and HYSPLIT trajectories are larger in December 2018 than in August 2022, please clarity?

*Response:*

*Thank you for your comment. The larger differences between ATLAS and HYSPLIT trajectories in December 2018 compared to August 2022 can be explained by the seasonal differences in the strength of upwelling and tropopause temperatures. During December, the upwelling is stronger, and the cold trap is colder. Under these conditions, HYSPLIT forward trajectories, which use kinematic vertical velocities, exhibit greater sensitivity to small perturbations or uncertainties in vertical motion. Schoeberl et al. (2011) pointed out that kinematic trajectories are more likely to encounter colder temperatures than those calculated using diabatic heating rates. This difference is reflected in our results, where HYSPLIT forward trajectories in December exhibit a more pronounced circling pattern within the TWP cold trap, whereas ATLAS trajectories mainly follow an eastward movement, with only one trajectory showing a circling pattern.*

*In contrast, during August, the cold trap is warmer with a weaker vertical temperature gradient, resulting in smaller differences between the two trajectory models. Despite these discrepancies, our seasonal conclusion remains robust: air masses show upward transport in winter and downward motion in summer based on the combined forward and backward trajectory analysis. Furthermore, as noted in the main text (L189), the strong upward motion observed in HYSPLIT's 20-day forward trajectories is likely an artificial signal, which does not affect our overall findings.*

*We have added the following explanation for clarity:*

*"As noted by Schoeberl et al. (2011), kinematic methods have a higher probability of encountering extremely low temperatures compared to diabatic methods, this may lead to overestimated*

*geopotential heights in HYSPLIT trajectories. This effect is evident in December when lower temperatures of the cold trap intensify temperature gradients. HYSPLIT trajectories therefore frequently exhibit circling patterns within the colder cold trap, whereas ATLAS shows only a single case of similar motion. In August, due to the warmer cold trap and weaker temperature gradients, differences between the two models are less pronounced."*

For figure A5: title "Forward trajectories in 13 Dec14:00" to "Forward trajectories on 13 Dec14:00"

*Corrected.*

---

## Editor Decision (ED1)

**Editor comments on egusphere-2024-3981**

*Evidence of Tropospheric Uplift into the Stratosphere via the Tropical Western Pacific Cold Trap* by Sun et al.

P2, L33: add "by" so that it reads "These cold temperatures significantly modify the pathways of air masses by dehydrating air and **by** the release of latent heat." Or change the latter part to "……..and releasing latent heat."

P2, L42: troposphere -> tropospheric

P3, L74-75: Add "used" so that it reads "Further details about the lidar measurements **used** in this study will be given in Sect. 2." However, I think this sentence is obsolete and should be deleted since almost the same is stated in the next paragraph.

P3, L76: Instead writing "In the following" I would suggest to write "In Sect. 2….".

P3, L78: delete "in Sect. 2".

P3, L81: "For an improved representation" not clear. Do you mean an improved representation of the results in this paper?

P3, L80-83: The text would be better readable if you would just write what is found in Section 3 without referring to every single subsection.

P3, L82-83: Rather write "In the discussion (Sect. 4) we provide …….and use this to discuss…….".

P3, L84-85: Either rephrase the sentence to be more clear or just write "In Sect. 5 conclusions are drawn."

P3, L90: add "measurements" or "observations" after "lidar" and replace "is" by "are".

P4, L93 and 97: Make one paragraph from these three paragraphs.

P4, L94: operating -> operations

P5, L132: Here and later ERA5 has been used, but ERA5 or any other reanalysis data has been described in the method section. Thus, at least one or two sentences plus references as for NCEP should be added here.

P5, L133: Remove "the" before "Appendix".

P5, L136: This is not clear. How can you derive information about condensation and dehydration from the cloud measurements? The presence of the clouds may be related to condensation, but not dehydration. Please rephrase the sentence.

P5, L138: Also here the sentence need to be revised. How can you just from the location of the trajectory in relation to the cloud top/bottom make statements on condensation or dehydration. I do not understand what physical mechanism should be behind this. Further, as your trajectory comparison shows differ the results between the trajectories significantly, thus, the location of the trajectory above or below the measured cloud could also be just related to model uncertainty.

P7, Figure 1: Space between panel c and d is missing.

P7, Figure 1 caption: remove space between "30" and degree sign.

P8, L178: Put "Fig. 2c" in parentheses.

P9, Figure 2: Remove "the" before "upper cloud" and "lower cloud" in the figure panels of Fig 2 a and b.

P9, Figure 2 caption: "as a function of time corresponds to Fig. 3a." not clear. Remove this text part.
Also remove "The backscatter ratio (BSR) at 532 nm as a function of time and altitude" since this sentence appears twice. The reference to Section 2 is not necessary here, I would suggest to omit this here.

P10, Figure 3 caption: put "Fig. 2" in parentheses and replace "in" by "for" and change "see supplement Fig. S1 and S2" to "see Fig S1 and Fig. S2". Further, I would suggest to write "for the winter case (Case 1)" (thus omitting "corresponding to").

P11, L229: remove space between opening parenthesis and "Fig."

P11, L239: will be -> can be found (?)

P11, L243: which case? Case 1 or Case 2. Generally, you could make more usage of the already introduced naming of cases to make it easier for the reader to follow.

P11, L247: Lidar -> lidar and replace "case" by "study" and change "previous studies in other regions" to "in previous studies over other regions".

P12, L252: you mean in an extremely cold "cold trap"?

P12, L253: add "is found" after analysis.

P12, Figure 4 caption: Add "Case 2" after "1 August 2022" and replace "in" by "for".

P13, Figure 5: Remove "the" before "upper cloud" and "lower cloud".

P13, Figure 5 caption: Replace "in" by "for" and add after case "Case 2" and delete "corresponding to Fig. 4".

P13, 261: Replace "Same to the" with "As for the"

P13, L261: This is not clear. Are you calculating here different trajectories or do you just consider another parameter along the trajectories?

P13, 263: Remove sentence "The trajectory setup details are described in Sect. 2.3 and Appendix A." Just add in parenthesis "see Sect. 2.3 and Appendix A".

P14, Figure 6 caption: "For the clarity of display, the trajectory points in the figure are sparsified at intervals of 24 points (24 h)." not clear. Does this mean you have an hourly output and only one point per day is plotted?

P14, Figure 6 caption: "For 20-d trajectories with similar results, please see the supplement Fig. S3." This sentence is also not clear and such a remark should rather appear in the main text than in the figure caption. Please rephrase or omit this sentence. I also do not really understand why you have two sets of trajectories, one 20d and one 10d?

P14, L280: Fig. 8 -> Fig. 7

P15, 283: ranges corresponding to -> ranges correspond to (?)

P15, L288: Remove parentheses around the references and add "are" after "but".

P15, L308-309: Delete sentence "The inter-hemispheric mixing controls the origins of air masses from the northern or southern hemisphere in the tropics." since it appears here for the second time.

P16, L316: compare Fig. 1 -> see Fig. 1

P16, L319: delete "in Fig. 9c" since Fig. 9c has just been referred to.

P16, L324: Tab. 1 -> Table 1

P17, L339: sinking -> descending

P18, L344-345: Delete "in a white dashed line". This should rather appear in the figure caption. However, in the figure I cannot see any dashed white line.

P18, L351: ……leading to dehydration of air parcels and cloud formation…. How do you achieve this? Cloud formation and dehydration are not measured. Please rephrase the text.

P18, L352: we -> was

P18, L368: What is a/the "triple La Nina"?

P18, L373: Sec. 3.4 -> Sect. 3.4

P18, L373: add references for the QBO phases.

P18, L378: Remove "(marked as a grey shaded area and dashed curves on the bottoms of Fig. 9a and b)"

P19, L399-400: Rephrase/Correct sentence "In consistency with other studies Randel and Park (2019); Pan et al. (2016), their work shows the possible pathway of air masses toward the TWP on the one hand."

P19, L405: Lidar -> lidar

P19, 407: What about the uncertainty of the trajectories? Couldn't it be that just the trajectories are not accurate enough?

Section 4: The discussions is extremely long, thus consider shortening.

P22, L477: Add references for the trajectory studies.

P23, Figure A1: Add altitudes for the upper and lower cloud.

P23, L481: Write "Case 2 (December)"

Put Fig. A1 and A2 on one page and Fig. A3 and A4 so that these can more easily be compared.

Appendix Figures: Remove "the" before upper cloud and lower cloud and add altitudes.

Figure A5: Also add here the altitude for the upper and lower cloud.

P26, L491: has been partly available -> has been partly made available (?)

P26, L496: Remove comma before Hersbach et al.

Supplement:
- Supplementary figures should be labelled as Fig. S1, Fig. S2 and so on.
- Remove in all figures "the" before upper cloud and lower cloud.
- Add the altitude for the upper and lower clouds.
- Replace "in" by "for" so that it reads "for the winter" add which Case you are referring to (Case 1 or Case 2).
- Don't repeat the entire caption text. Write "As Fig. xx, but……" and point out the differences to the other figure.

- Fig. S3 caption: What is meant with simulation results? Once you calculate 10d trajectories and once 20d trajectories. Why you do this did not become clear while reading your manuscript.

---

## Author Response (AR2)

Reply to editor comments on egusphere-2024-3981

Evidence of Tropospheric Uplift into the Stratosphere via the Tropical Western Pacific Cold Trap by Sun et al.

P2, L33: add "by" so that it reads "These cold temperatures significantly modify the pathways of air masses by dehydrating air and by the release of latent heat." Or change the latter part to "……..and releasing latent heat."

*Added.*

P2, L42: troposphere -> tropospheric

*Corrected.*

P3, L74-75: Add "used" so that it reads "Further details about the lidar measurements used in this study will be given in Sect. 2." However, I think this sentence is obsolete and should be deleted since almost the same is stated in the next paragraph.

*Thanks, we've deleted this sentence.*

P3, L76: Instead writing "In the following" I would suggest to write "In Sect. 2….".

*Corrected.*

P3, L78: delete "in Sect. 2".

*Deleted.*

P3, L81: "For an improved representation" not clear. Do you mean an improved representation of the results in this paper?

*Deleted.*

P3, L80-83: The text would be better readable if you would just write what is found in Section 3 without referring to every single subsection.

*Corrected.*

P3, L82-83: Rather write "In the discussion (Sect. 4) we provide …….and use this to discuss…….".

*Corrected.*

P3, L84-85: Either rephrase the sentence to be more clear or just write "In Sect. 5 conclusions are drawn."

*Corrected.*

P3, L90: add "measurements" or "observations" after "lidar" and replace "is" by "are".

*Corrected.*

P4, L93 and 97: Make one paragraph from these three paragraphs.

*Corrected.*

P4, L94: operating -> operations

*Corrected.*

P5, L132: Here and later ERA5 has been used, but ERA5 or any other reanalysis data has been described in the method section. Thus, at least one or two sentences plus references as for NCEP should be added here.

*Thanks, we've added the references here.*

P5, L133: Remove "the" before "Appendix".

*Removed.*

P5, L136: This is not clear. How can you derive information about condensation and dehydration from the cloud measurements? The presence of the clouds may be related to condensation, but not dehydration. Please rephrase the sentence.
*Reply:*

*Thank you for pointing out this confusion. Indeed, the lidar cloud measurements alone only directly indicate the presence of condensation processes, as clouds form by condensation. Dehydration, however, is inferred indirectly because cloud formation implies that water vapor has condensed into ice particles, thus potentially removing moisture from the air parcel. We acknowledge that lidar data alone do not directly measure dehydration. We will rephrase this sentence clearly to specify that dehydration is an inferred rather than directly measured process.*

*We rephrased this sentence in L136:*

*"When these clouds are detected, it suggests that condensation processes have occurred, potentially leading to associated dehydration of air masses through the removal of water vapor by ice particle formation."*

P5, L138: Also here the sentence need to be revised. How can you just from the location of the trajectory in relation to the cloud top/bottom make statements on condensation or dehydration. I do not understand what physical mechanism should be behind this. Further, as your trajectory comparison shows differ the results between the trajectories significantly, thus, the location of the trajectory above or below the measured cloud could also be just related to model uncertainty.

*Reply:*

*We agree with your comment that the vertical displacement of trajectory points relative to the cloud layers does not directly provide evidence of dehydration processes. Our intended meaning was to use the vertical displacement observed from the trajectory model as an indicator of general vertical motion (uplift or descent) of air masses after clouds are detected.*

*We will clarify in the manuscript that vertical displacement inferred from trajectories serves only as an indicator of possible vertical transport pathways, and explicitly acknowledge model uncertainties. For the uncertainties, we compare two trajectory models with different reanalysis data and vertical cordite approaches, as shown in the appendix. We also have rephrased these sentences to illustrate our concern about uncertainty by two trajectory comparisons:*

*"…When considering potential uncertainties arising from trajectory models, we compared calculations using these two models. Although the comparisons revealed differences and inherent uncertainties (See Appendix A), they show a consistent circulation pattern: air masses move upward/downward toward the cloud layer from lower altitudes in backward trajectories, and subsequently move upward/downward away from the cloud layer toward higher altitudes in forward trajectories. This pattern supports the inference of an ascending/descending transport process."*

P7, Figure 1: Space between panel c and d is missing.

*Corrected and space between panels was added.*

P7, Figure 1 caption: remove space between "30" and degree sign.

*Corrected.*

P8, L178: Put "Fig. 2c" in parentheses.

*Corrected.*

P9, Figure 2: Remove "the" before "upper cloud" and "lower cloud" in the figure panels of Fig 2 a and b.

*Removed.*

P9, Figure 2 caption: "as a function of time corresponds to Fig. 3a." not clear. Remove this text part.

Also remove "The backscatter ratio (BSR) at 532 nm as a function of time and altitude" since this sentence appears twice. The reference to Section 2 is not necessary here, I would suggest to omit this here.

*Corrected.*

P10, Figure 3 caption: put "Fig. 2" in parentheses and replace "in" by "for" and change "see supplement Fig. S1 and S2" to "see Fig S1 and Fig. S2". Further, I would suggest to write "for the winter case (Case 1)" (thus omitting "corresponding to").

*Corrected.*

P11, L229: remove space between opening parenthesis and "Fig."

*Corrected.*

P11, L239: will be -> can be found (?)

*Corrected.*

P11, L243: which case? Case 1 or Case 2. Generally, you could make more usage of the already introduced naming of cases to make it easier for the reader to follow.

*Corrected.*

*The same phrases for the "case" were corrected in the following draft.*

P11, L247: Lidar -> lidar and replace "case" by "study" and change "previous studies in other regions" to "in previous studies over other regions".

*Corrected.*

P12, L252: you mean in an extremely cold "cold trap"?

*Yes, we mean an extremely cold "cold trap." Specifically, we refer to conditions characterized by extremely low tropopause temperatures, which enhance dehydration processes and thus strengthen upward transport over the TWP during NH winter. We will clarify this in the revised text.*
*We rephrased it as:*

*"This contrast aligns with … supporting enhanced upwelling over the TWP under conditions of an extremely cold 'cold trap' ..."*

P12, L253: add "is found" after analysis.
*Added.*

P12, Figure 4 caption: Add "Case 2" after "1 August 2022" and replace "in" by "for".

*Added.*

P13, Figure 5: Remove "the" before "upper cloud" and "lower cloud".

*Removed.*

P13, Figure 5 caption: Replace "in" by "for" and add after case "Case 2" and delete "corresponding to Fig. 4".

*Corrected.*

P13, 261: Replace "Same to the" with "As for the"

*Corrected.*

P13, L261: This is not clear. Are you calculating here different trajectories or do you just consider another parameter along the trajectories?

*Reply:*

*We did not perform separate trajectory calculations for potential temperature. Instead, we computed potential temperature as an additional meteorological parameter along the previously calculated trajectories. Thus, the trajectories themselves remain the same; potential temperature is presented to illustrate the quasi-horizontal nature of transport over the TWP.*

*Rephrased as:*

*"Additionally, we present potential temperature as an additional meteorological parameter along the previously calculated trajectories to illustrate the quasi-horizontal transport over the TWP."*

P13, 263: Remove sentence "The trajectory setup details are described in Sect. 2.3 and Appendix A." Just add in parenthesis "see Sect. 2.3 and Appendix A".

*Corrected.*

P14, Figure 6 caption: "For the clarity of display, the trajectory points in the figure are sparsified at intervals of 24 points (24 h)." not clear. Does this mean you have an hourly output and only one point per day is plotted?

*Yes, for the clarity and not overwhelming the plot with 1-h interval output. We cut this sentence in the caption. And added the sentence in L264:*

*"Although trajectories were calculated hourly, points are plotted here at daily intervals (every 24 hours) to enhance visual clarity. All trajectories were originally calculated over 20 days, but only the first 10 days are presented here to avoid overwhelming the figure; full 20-day trajectories are shown in Fig. S3 in the supplement."*

P14, Figure 6 caption: "For 20-d trajectories with similar results, please see the supplement Fig. S3." This sentence is also not clear and such a remark should rather appear in the main text than in the figure caption. Please rephrase or omit this sentence. I also do not really understand why you have two sets of trajectories, one 20d and one 10d?

*Reply:*

*Here we want to present 10-day backward and forward trajectory results of air masses corresponding to Fig. S3 in the supplement. While all trajectory calculations in this study were performed over 20 days, only 10-day trajectories are shown in Fig. 6 to maintain visual clarity. Here, the complete 20-day trajectories are presented to provide comprehensive information without overwhelming the main manuscript.*

P14, L280: Fig. 8 -> Fig. 7

*Corrected.*

P15, 283: ranges corresponding to -> ranges correspond to (?)

*Corrected.*

P15, L288: Remove parentheses around the references and add "are" after "but".

*Corrected.*

P15, L308-309: Delete sentence "The inter-hemispheric mixing controls the origins of air masses from the northern or southern hemisphere in the tropics." since it appears here for the second time.

*Corrected.*

P16, L316: compare Fig. 1 -> see Fig. 1

*Corrected.*

P16, L319: delete "in Fig. 9c" since Fig. 9c has just been referred to.

*Corrected.*

P16, L324: Tab. 1 -> Table 1

*Corrected.*

P17, L339: sinking -> descending

*Corrected.*

P18, L344-345: Delete "in a white dashed line". This should rather appear in the figure caption. However, in the figure I cannot see any dashed white line.

*Deleted.*

P18, L351: ……leading to dehydration of air parcels and cloud formation…. How do you achieve this? Cloud formation and dehydration are not measured. Please rephrase the text.

*Rephrased:*

*"… cloud formation …, as inferred from our lidar observations of cirrus clouds and relative humidity profiles in December."*

P18, L352: we -> was

*Corrected.*

P18, L368: What is a/the "triple La Nina"?

*Cut triple.*

P18, L373: Sec. 3.4 -> Sect. 3.4

*Corrected.*

P18, L373: add references for the QBO phases.

*Added.*

P18, L378: Remove "(marked as a grey shaded area and dashed curves on the bottoms of Fig. 9a and b)"

*Corrected.*

P19, L399-400: Rephrase/Correct sentence "In consistency with other studies Randel and Park (2019); Pan et al. (2016), their work shows the possible pathway of air masses toward the TWP on the one hand."

*Reply:*

*Corrected as:*

*Honomichl and Pan (2020), using ERA-Interim data from 1979–2017, analyzed transport pathways from the ASM via the western Pacific anticyclone to the western Pacific, consistent with other studies (Randel et al., 2019; Pan et al., 2016), highlighting potential air mass transport routes toward the TWP.*

P19, L405: Lidar -> lidar

Corrected.

P19, 407: What about the uncertainty of the trajectories? Couldn't it be that just the trajectories are not accurate enough?

*Reply:*

*Thank you very much for your questions. Indeed, trajectory calculations inherently contain uncertainties and may not always be fully accurate. To evaluate and address this uncertainty, we performed trajectory analyses using both the ATLAS and HYSPLIT models and compared their outputs. As discussed in detail in the Appendix, we quantified differences between these two trajectory models and examined the sources of discrepancies. Our analysis indicates that the primary cause of uncertainty between ATLAS and HYSPLIT trajectories arises from differences in their representation of the vertical coordinate. Importantly, artificial effects related to the kinematic vertical velocity calculations primarily occur only towards the very end of the 20-day trajectories. Apart from this, both models consistently show the same seasonal transport pattern: ascending air masses in winter and descending in summer, providing confidence in our conclusions despite these acknowledged uncertainties.*

*We added the following sentence in L431 for the uncertainty of the trajectories:*

*"To account for the uncertainty inherent in trajectory calculations, we compared trajectory results from two Lagrangian transport models, ATLAS and HYSPLIT, confirming that both models consistently show the same seasonal transport patterns, with discrepancies primarily arising from differences in vertical coordinate."*

Section 4: The discussions is extremely long, thus consider shortening.

We split the discussion section into two sections, one is a subsection in the result section for the different transport pathways (With Fig. 9) and another is the discussion section.

P22, L477: Add references for the trajectory studies.

*Corrected.*

P23, Figure A1: Add altitudes for the upper and lower cloud.

*Corrected.*

P23, L481: Write "Case 2 (December)"

*Corrected.*

Put Fig. A1 and A2 on one page and Fig. A3 and A4 so that these can more easily be compared.

*Corrected.*

Appendix Figures: Remove "the" before upper cloud and lower cloud and add altitudes.

*Corrected.*

Figure A5: Also add here the altitude for the upper and lower cloud.

*Added.*

P26, L491: has been partly available -> has been partly made available (?)

*Corrected.*

P26, L496: Remove comma before Hersbach et al.

*Corrected.*

Supplement:

• Supplementary figures should be labelled as Fig. S1, Fig. S2 and so on.

*Corrected.*

• Remove in all figures "the" before upper cloud and lower cloud.

*Corrected.*

• Add the altitude for the upper and lower clouds.

*Added.*

• Replace "in" by "for" so that it reads "for the winter" add which Case you are referring to (Case 1 or Case 2).

*Corrected.*

• Don't repeat the entire caption text. Write "As Fig. xx, but……" and point out the differences to the other figure.

*Corrected.*

• Fig. S3 caption: What is meant with simulation results? Once you calculate 10d trajectories and once 20d trajectories. Why you do this did not become clear while reading your manuscript.

*Reply:*

*Here we want to present 20-day backward and forward trajectory results of air masses corresponding to Fig. 6 in the main manuscript. While all trajectory calculations in this study were performed over 20 days, only 10-day trajectories are shown in Fig. 6 to maintain visual clarity. Here, the complete 20-day trajectories are presented to provide comprehensive information without overwhelming the main manuscript.*

*Best regards,*

*Co-authors*

---

## Editor Decision (ED2)

**Editor comments on 2nd revision**

**Specific comments:**

P1, L7: The statement: "Latent heat released during cloud formation drives the ascent of air masses." is not correct. I guess you mean that the latent heat release causes a cooling that can lead to further ascending of air masses. However, if an air mass ascends or not is driven by the stability of the atmosphere. Latent heating has (if there is an impact) only a minor impact.

P1, L8: "A case study in December" -> could that be rephrased as well. You are doing a case study, but the measurements are available not only for December 2018.

P2, L32-33: "These cold temperatures significantly modify the pathways of air masses by dehydrating air and by the release of latent heat" -> they do not modify the pathways it is rather the amount of clouds or cirrus that are formed.

P2, L34 and throughout the manuscript: The term "dehydration" is misused. Dehydration is usually caused by sedimenting ice particles. Either you have to write this more clearly or you should omit the term dehydration here and at some other places in the manuscript. Note, while reading your paper and checking again on this processes in the TTL, I realized that the term that is missing here is "freeze-drying". This is the process that causes the dehydration in the TTL (see e.g. Uemura et al., https://doi.org/10.1002/2013JD021381, 2014).

P3, L58: here you should replace "dehydration" rather by cloud formation processes. Using CALIOP cloud observations you observe when and where clouds occurred and in combination with models you are able to investigate the formation processes. You could also write "cloud formation and related processes as dehydration".

P5, L124: "………potentially leading to associated dehydration of air masses through the removal of water vapor by ice particle formation." Add "and subsequent sedimentation of the ice particles" since there will be no dehydration of the air mass if the ice particles haven't sedimented. Note, same here as for P2, L34 while reading your paper and checking again on this processes in the TTL, I realized that the term that is missing here is "freeze-drying". This is the process that causes the dehydration in the TTL (see e.g. Uemura et al., https://doi.org/10.1002/2013JD021381, 2014) .

P5, L129-130: "………..with a hybrid vertical coordinate within the TTL changing from kinematic to diabatic." I hope you compare the kinematic versions of the model and use the diabatic version for further analysis. Could you rephrase the sentence as well to be more precise which versions of the models you actually compare?

P8, L167: "….air mass to release latent heat and subsequently rise into the stratosphere" you mean from deep convective clouds? Here of course the latent heat may have a higher impact and may have an influence on the ascent of air masses.

P8, L167-169: "This supersaturated and cold environment near the CPT suggests a condition that led to further dehydration of the air mass flow in the altitude range above 14 km up to the CPT." If the process of freeze-drying is explained somewhere before in the manuscript, this sentence may ok as it is now, but otherwise it should be improved as well.

P9, L195-196: "We have compared and validated the HYSPLIT results with those of ATLAS shown in Appendix A." This sentence is obsolete since you have mentioned this already several times before

and a few lines later you write again "The comparison between the two model results is described in detail in Appendix A." This paragraph should be improved.

P11, L236: "………dispersion of the vertical coordinates of stratospheric kinematics." Sentence not clear, please rephrase.

P11, L237: "The comparison details between the results of the two models are in Appendix A." Obsolete here since this has been mentioned several times, just add "see Appendix A" in parenthesis.

P18, L378-379: " …. bringing high concentrations of species related to human activities, such as $O_3$, CO, and $NO_x$." Text part uncomplete. Add something like " to this/other region". Not clear what you actually mean here, if the pollutants are coming to Southeast Asia or transported away from there.

P19, L398: What is meant with "consistent with other studies"? Same type of study or deriving the same/similar results? Please rephrase the sentence.

P19, L415: "Latent heat can be released due to the condensation of water vapor, resulting in the dominant ascent of air in this region." Please improve this sentence and make clear that you mean latent heat release in connection with deep convection.

P22, L455-456: "All trajectories within a cirrus cloud are initiated simultaneously every hour during the time of the lidar observation." Do you really start every hour new trajectories or do you use an hourly output for the trajectories? Please add this information as well. In the former case, over how many hours are trajectories started and how many trajectories have you calculated in total? In the latter case, please rephrase the sentence to be more clear.

P22, L456ff: You are comparing two trajectory models with different vertical coordinates for trajectory calculation. However, both consider diabatic processes. Nevertheless, due to the different approaches differences occur. What I miss in your manuscript is a clear statement what you want to achieve with comparing these two models. Do you want to investigate which method produced the more reliable results? Since the calculations are also based on different reanalysis this makes an assessment quite difficult.

**Technical corrections:**

P9, L186: cold traps -> cold trap

P9, Figure 2 caption: Add "for" so that it reads "Lidar and radiosonde observations **for** a typical case of cirrus cloud measurements on 13 December 2018."

P9, Figure 2 caption: remove second full stop after "marked" and Section 2 -> Sect 2.

P11, L218 : "this case" -> which case? Please be more precise.

P11, L240: "Case 2" a bit lost year. Either put this in parentheses with the reference to Fig 4 or include this in the sentence.

P11,L248: Since you mean "an extremely cold 'cold trap' you should write it also like this.

P13, L259: "capture variations" variations of what? Please be more precise.

P13, L259: "Although trajectories were calculated hourly" Not the trajectories are calculated hourly, you calculate these with an hourly output time step. Please rephrase the sentence to be more clear.

P14, Figure 6 caption: cloud layer -> cirrus cloud layer

P14, L287: Remove parenthesis around the reference of "Sun et al., 2023a".

P15, L295: "the ascending air masses higher than the 400 K level account……" you mean the air masses "that" ascend higher than the 400 K level? Please correct the sentence.

P15, L299: "…….., enhancing tropical stratospheric entry." This sentence part feels uncomplete.

P18, L368: dynamics -> dynamical processes

P18, L373: Research -> Recent research studies ……….(e.g.  Rao et al.)  if there are several studies or A recent research study ………(Rao et al.) if this is only discussed in Rao et al.

P18, L383:  vast -> west (?)

---

## Author Response (AR3)

Response to Editor comments on 2nd revision

We sincerely appreciate your careful review and the time you took to read our manuscript again. Your insightful comments and suggestions have helped us improve the clarity and accuracy of the paper. We have carefully addressed each of your comments, and we believe the revised version is now clearer and more robust.

*Specific comments:*

*P1, L7: The statement: "Latent heat released during cloud formation drives the ascent of air*

*masses." is not correct. I guess you mean that the latent heat release causes a cooling that can lead to further ascending of air masses. However, if an air mass ascends or not is driven by the stability of the atmosphere. Latent heating has (if there is an impact) only a minor impact.*

*P1, L8: "A case study in December" -> could that be rephrased as well. You are doing a case study, but the measurements are available not only for December 2018.*

**Response to comments to abstract:**

Thank you very much for your thoughtful comments and questions. Please find our responses regarding the abstract below:

**1. Regarding latent heat and air mass ascent (P1, L7):**

Thank you for highlighting this important point. We agree with your suggestion that latent heat release alone does not directly drive the ascent of air masses. To avoid any misunderstanding, we have removed this statement from the abstract. We revised the sentence as follows:

"These conditions lead to water vapor condensation, forming thin cirrus clouds which can be measured as an indicator of the ascent of air masses".

**2. Regarding the wording "A case study in December" (P1, L8):**

Thank you for pointing out the ambiguity here. We agree this wording can imply measurements are limited to December 2018, which is not the case. We revised this sentence for clarity as follows:

Revised sentence: "A representative example from December 2018 shows a subvisible cirrus cloud layer (optical depth < 0.03) measured by lidar, coinciding with high supersaturation observed by radiosonde."

*P2, L32-33: "These cold temperatures significantly modify the pathways of air masses by dehydrating air and by the release of latent heat" -> they do not modify the pathways it is rather the amount of clouds or cirrus that are formed.*

*P2, L34 and throughout the manuscript: The term "dehydration" is misused. Dehydration is usually caused by sedimenting ice particles. Either you have to write this more clearly or you should omit the term dehydration here and at some other places in the manuscript. Note, while reading your paper and checking again on this processes in the TTL, I realized that the term that is missing here is "freeze-drying". This is the process that causes the dehydration in the TTL (see e.g. Uemura et al., https://doi.org/10.1002/2013JD021381, 2014).*

**Response to P2, L32-34 and the following paragraph:**

**1. Regarding the term "modify pathways" (P2, L32-33):**

We thank the reviewer for this insightful comment. We agree that the phrase "modify the pathways" is not precise, as the cold temperatures primarily influence the formation and persistence of cirrus clouds rather than directly altering air mass pathways. We have revised the sentence to clarify:

Revised sentence (P2, L32-33): "These cold temperatures significantly influence the formation of cirrus clouds and thus the humidity in TTL."

**2. Regarding the term "dehydration" (P2, L34):**

We thank the reviewer for this insightful comment. We agree that the term *"dehydration"* should be used with greater precision, particularly to emphasize the underlying microphysical processes. In the revised manuscript, we clarify the cloud formation via the *freeze-drying* process and that it is related to dehydration in the TTL. Air slowly ascending through the cold tropical tropopause forms ice particles that may sediment out of the airmass, reducing its water vapor content. Thank you for providing the literature by Ueyama et al. (2014), which stated that "air is dehydrated via the 'freeze-drying' process."

We removed the sentence (P2, L34-35) where we misuse the term "dehydration".

We have clarified this "freeze-drying" process in our revised manuscript accordingly (P2, L34):

"As air masses move horizontally within the TTL for long distances, they slow ascent, experiencing the coldest temperatures. Moisture is removed via condensation in a freeze-drying and subsequently sedimentation process (Ueyama et al., 2014)."

We have checked the paragraph accordingly (P2, L30-40) to keep this paragraph clearly stating the terms "freezing-drying" and "dehydration". We also checked for misuse throughout the manuscript to ensure that "dehydration" is used in the context of the freeze-drying mechanism. We appreciate the reviewer's guidance in helping us improve the clarity and accuracy of our terminology.

*P3, L58: here you should replace "dehydration" rather by cloud formation processes. Using CALIOP cloud observations you observe when and where clouds occurred and in combination with models you are able to investigate the formation processes. You could also write "cloud formation and related processes as dehydration".*

**Response:**

Thanks. We replaced the "***dehydration***" with "***cloud formation and related dehydration processes,***" as you suggested.

*P5, L124: "………potentially leading to associated dehydration of air masses through the removal of water vapor by ice particle formation." Add "and subsequent sedimentation of the ice particles" since there will be no dehydration of the air mass if the ice particles haven't sedimented. Note, same here as for P2, L34 while reading your paper and checking again on this processes in the TTL, I realized that the term that is missing here is "freeze-drying". This is the process that causes the dehydration in the TTL (see e.g. Uemura et al., https://doi.org/10.1002/2013JD021381, 2014).*

**Response:**

Thanks, we added "and their subsequent sedimentation."

*P5, L129-130: "………..with a hybrid vertical coordinate within the TTL changing from kinematic to diabatic." I hope you compare the kinematic versions of the model and use the diabatic version for further analysis. Could you rephrase the sentence as well to be more precise which versions of the models you actually compare?*

**Response:**

We thank the reviewer for this important observation. In our study, the HYSPLIT trajectories were driven by the GDAS1 meteorological dataset using a kinematic vertical coordinate, while the ATLAS trajectories were driven by ERA5 reanalysis using a hybrid vertical coordinate that gradually changes from kinematic to fully diabatic above 100 hPa. So, we admit that the comparison between HYSPLIT and ATLAS reflects the combined influence of differences in reanalysis data, trajectory model, and vertical coordinate approach.

Our intention was not the thorough inter-comparison of models and to isolate the effect of vertical coordinate alone but rather simply to demonstrate the sensitivity of TTL trajectory pathways to different common modeling frameworks and inputs. We suggest clarifying this and revising the sentence (L129-130) in the manuscript to more precisely reflect this scope.

In a later specific comment in this response, we continue our answer to this question with more detail, also in the context of the Appendix.

**Revised sentences (P5, L129-130):**

"To investigate the sensitivity of trajectory pathways calculated with HYSPLIT setup, we conducted additional simulations using the Alfred-Wegener-InsTitute LAgrangian Chemistry/Transport System (ATLAS) transport model (Wohltmann and Rex, 2009), driven by ERA5 reanalysis data and applying a hybrid vertical coordinate that gradually changes from kinematic to fully diabatic above 100 hPa."

We suggest adding the following sentence (P5, L130) to clearly state the aim of using two models:

"In this study, a thorough inter-comparison between the two models is not the focus. Instead, the relative distributions of different pathways are of main interest. In this sense, the example case study results between the two models are given, and the differences in reanalyzing data inputs are neglected."

*P8, L167: "….air mass to release latent heat and subsequently rise into the stratosphere" you mean from deep convective clouds? Here of course the latent heat may have a higher impact and may have an influence on the ascent of air masses.*

**Response:**

Thank you for your comment. We did not intend to refer to deep convective clouds here. Instead, we are describing the large-scale ascent of air masses into the stratosphere. As the air ascends and passes through the cold trap near the tropopause, water vapor tends to condense out. This condensation is not the cause of ascent but rather a consequence of the cooling during ascent, and it serves as an indicator of the vertical transport.

We revised the sentence (P8, L167) as follows: "…the low-temperature conditions cause water vapor to condense and form cirrus clouds, which in turn indicate vertical transport into the stratosphere."

*P8, L167-169: "This supersaturated and cold environment near the CPT suggests a condition that led to further dehydration of the air mass flow in the altitude range above 14 km up to the CPT." If the process of freeze-drying is explained somewhere before in the manuscript, this sentence may ok as it is now, but otherwise it should be improved as well.*

**Response:**

Thank you for the suggestion. We revised the sentence to clarify the freeze-drying process in this context, specifically referring to cirrus cloud formation and ice particle sedimentation as the mechanism for further dehydration near the CPT.

We revised the sentence (P8, L167-169) as follows**: "***This supersaturated and cold environment near the CPT suggests a condition favorable for cirrus cloud formation and further dehydration of the air mass via ice particle sedimentation in the altitude range above 14 km up to the CPT.***"**

*P9, L195-196: "We have compared and validated the HYSPLIT results with those of ATLAS shown in*

*Appendix A." This sentence is obsolete since you have mentioned this already several times before and a few lines later you write again "The comparison between the two model results is described in detail in Appendix A." This paragraph should be improved.*

**Response:**

Thank you for the comment. We revised the paragraph (P9, L195-198) to remove the redundancy and improve clarity:

"We compared and validated the HYSPLIT results with those of ATLAS. In comparison, ATLAS trajectories do not show this extreme uplift. Otherwise, the results of the two models are comparably consistent, especially regarding the NH winter air uplift. A detailed comparison between the two model results is described in Appendix A."

*P11, L236: "………dispersion of the vertical coordinates of stratospheric kinematics." Sentence not clear, please rephrase.*

**Response:**

We revised the sentence (P11, L236) for clarity as follows: "… dispersion in stratospheric trajectory calculations."

*P11, L237: "The comparison details between the results of the two models are in Appendix A."*

*Obsolete here since this has been mentioned several times, just add "see Appendix A" in parenthesis.*

**Response:**

We removed this sentence (P11, 237).

*P18, L378-379:" …. bringing high concentrations of species related to human activities, such as O3,*

*CO." Text part uncomplete. Add something like "to this/other region". Not clear what you actually mean here, if the pollutants are coming to Southeast Asia or transported away from there.*

**Response:**

Thank you for the suggestion. We revised the sentence to clarify that the pollutants are transported from Southeast Asia into the TWP.

"During NH winter, the dominant air mass origins are within the NH (red arrows and circles in Fig. 9, specifically Southeast Asia, transporting high concentrations of species related to human activities, such as $O_3$ and CO to the TWP."

*P19, L398: What is meant with "consistent with other studies"? Same type of study or deriving the same/similar results? Please rephrase the sentence.*

**Response:**

Thank you for the comment. We revised the sentence to clarify that the studies cited derive similar results regarding air mass transport pathways toward the TWP.

We rephrased the sentences (P19, L398) as follows: "Similar results from other studies (e.g., Randel et al., 2019, Pan et al., 2016) also highlight potential air mass transport routes toward the TWP."

*P19, L415: "Latent heat can be released due to the condensation of water vapor, resulting in the dominant ascent of air in this region." Please improve this sentence and make clear that you mean latent heat release in connection with deep convection.*

**Response:**

Thank you for the helpful comment. We revised the sentence to improve clarity and better reflect the physical processes involved. In this section, our intention is to describe the condensation of water vapor as a result of large-scale uplift and cooling rather than to suggest that latent heat release is the primary driver of the convective ascent. Since the sentence should focus on large-scale transport during winter, we have revised the sentence (P19, L415) to improve the clarity as follows:

"Latent heat can be released because of the condensation of water vapor, which occurs as a result of the dominant ascent and extremely low temperatures in this region."

*P22, L455-456: "All trajectories within a cirrus cloud are initiated simultaneously every hour during the time of the lidar observation." Do you really start every hour new trajectories or do you use an hourly output for the trajectories? Please add this information as well. In the former case, over how many hours are trajectories started and how many trajectories have you calculated in total? In the latter case, please rephrase the sentence to be more clear.*

*P22, L456ff: You are comparing two trajectory models with different vertical coordinates for trajectory calculation. However, both consider diabatic processes. Nevertheless, due to the different approaches differences occur. What I miss in your manuscript is a clear statement what you want to achieve with comparing these two models. Do you want to investigate which method produced the more reliable results? Since the calculations are also based on different reanalysis this makes an assessment quite difficult.*

**Response to these questions regarding P22 L455-456 and 456ff:**

**1. Regarding initiation of trajectories (P22, L455-456):**

We appreciate your request for clarification. To clarify, trajectories are indeed initialized every hour independently during periods when the cirrus cloud is continuously observed to persist for more than one hour. Trajectories are initiated simultaneously every hour in time within the detected cirrus cloud, vertically spaced at 100 m intervals in the output vertical coordinate of the lidar measurement and converted to pressure levels using the closest radiosonde data in time. The total number of trajectories calculated depends on the duration of cloud presence, with each hour producing an independent set of trajectories. To reflect this clearly, we revise the sentence in P21, L454 as follows:

"Trajectory starting points within a cirrus cloud layer are vertically spaced at 100 m intervals in geometric height and converted to pressure levels using radiosonde data from the PWS, taken closest in time to the cirrus cloud observation. All trajectories within a cirrus cloud are initiated simultaneously every hour during the time of the lidar observation and with an output time step of 1 h."

We suggested adding an explicit statement of the number of trajectories we initialized in this paragraph:

"For a detected cloud layer case, the number of trajectories is thus the cloud thickness divided by 100m times the durations of the cloud observation in hour."

**2. Purpose of comparing two trajectory models (P22, L456ff):**

Thank you for emphasizing the importance of clearly stating our objectives when comparing these two trajectory models. The inter-comparison of the ATLAS-HYSPLIT models is not our intention. Instead, we acknowledge that there have been discussions about these two vertical velocity approaches, and we want to justify the sensitivity of the trajectory calculation by HYSPLIT through the simple case study comparisons.

We revised sentences (P22, L443) to state the aim of using the additional model clearly as follows:

"To investigate the sensitivity of trajectory pathways calculated with HYSPLIT setup, we use the additional …"

However, we must clarify that a detailed comparison between different Lagrangian models is beyond our scope due to significant computational and practical constraints. We could theoretically extend our comparison of HYSPLIT and ATLAS by using the same reanalysis dataset, but this would require more time and computational effort than we originally intended for the paper. We suggest clarifying this in the paper on P22, L470:

"In the scope of this study, the inter-comparison between the two models was not the focus. The examples of case study simulations are given here to show the similar relative distribution of pathways over the TWP, which is the focus of this study. In this sense, differences in reanalysis data inputs were neglected when comparing the sensitivity of trajectory calculations through case studies. "

It is not our intention to establish the superiority of one model over the other. There are discussions about the performances of these two velocity approaches (e.g., Honomichl and Pan 2020), but we do not aim to solve this. With our additional case study using the ATLAS setup, we contributed to this discussion. Our comparison suggests that the different vertical velocity approaches are important to the study of long-time transport in UTLS, and further study is needed.

We revised sentences (P22, L484): "Both models provide evidence of the predominance of ascending air masses over Palau in Case 1 (the winter case) ...... A thorough model inter-comparison accounting for differences in both vertical velocity and reanalysis datasets would further clarify remaining uncertainties and add new insights to the discussion of transport studies in the TTL (Honomichl and Pan 2020)."

We thank you once again for these important suggestions, which have significantly improved the clarity of our manuscript.

*Technical corrections:*

*P9, L186: cold traps -> cold trap*

Corrected.

*P9, Figure 2 caption: Add "for" so that it reads "Lidar and radiosonde observations for a typical case of cirrus cloud measurements on 13 December 2018."*

Corrected.

*P9, Figure 2 caption: remove second full stop after "marked" and Section 2 -> Sect 2.*

Corrected.

*P11, L218: "this case" -> which case? Please be more precise.*

*Corrected as: "Case 2 (summer)".*

*P11, L240: "Case 2" a bit lost year. Either put this in parentheses with the reference to Fig 4 or include this in the sentence.*

*Corrected as: "(Case 2, Fig. 4b)".*

*P11, L248: Since you mean "an extremely cold 'cold trap' you should write it also like this.*

Corrected.

*P13, L259: "capture variations" variations of what? Please be more precise.*

Corrected as: "capture variations in time and vertical positions of the air masses initialized within the cloud layers."

*P13, L259: "Although trajectories were calculated hourly" Not the trajectories are calculated hourly, you calculate these with an hourly output time step. Please rephrase the sentence to be more clear.*

Corrected as: "Although the output time step of the trajectory is 1 hour".

*P14, Figure 6 caption: cloud layer -> cirrus cloud layer*

Corrected.

*P14, L287: Remove parenthesis around the reference of "Sun et al., 2023a".*

Corrected.

*P15, L295: "the ascending air masses higher than the 400 K level account……" you mean the air masses "that" ascend higher than the 400 K level? Please correct the sentence.*

Corrected.

*P15, L299: "…….., enhancing tropical stratospheric entry." This sentence part feels uncomplete.*

Corrected as: "enhancing the entry of tropical air masses into the stratosphere."

*P18, L368: dynamics -> dynamical processes*

Corrected.

*P18, L373: Research -> Recent research studies ………. (e.g. Rao et al.) if there are several studies or*

*A recent research study ……… (Rao et al.) if this is only discussed in Rao et al.*

Corrected.

*P18, L383: vast -> west (?)*

Corrected.